# Conditional Risk-Averse Constrained Reinforcement Learning

**James McCarthy**                                                   *mccarj21@tcd.ie*
*IBM Research, Ireland*
*Trinity College Dublin*

**Radu Marinescu**                                        *radu.marinescu@ie.ibm.com*
*IBM Research, Ireland*

**Elizabeth Daly**                                        *elizabeth.daly@ie.ibm.com*
*IBM Research, Ireland*

**Ivana Dusparic**                                            *ivana.dusparic@tcd.ie*
*Trinity College Dublin*

**Reviewed on OpenReview:** *https://openreview.net/forum?id=JJFFx1HVHi*

## Abstract

In Risk-averse Constrained Reinforcement Learning (RaCRL), the optimal tolerance for risk often depends on a preference over the trade-off between reward and safety. This trade-off is influenced by environmental uncertainty, which is generally difficult to quantify, in turn making its effect on an agent's performance difficult to predict at the outset of training. Conventional RaCRL approaches typically train agents under a fixed risk level, set at the beginning of training, leading to an agent with a fixed, often conservative, reward-safety trade-off at deployment time. In this paper, we introduce Conditional Risk-averse Actor Critic (CRAC), a novel algorithm for RaCRL that conditions the agent on risk levels sampled during both exploration and learning. Through exploring and learning from diverse experiences across varied risk levels, CRAC generalises effectively across a spectrum of risk preferences, enabling the deployment of a single agent at risk levels chosen by a user. We evaluate CRAC across a set of environments with increasing difficulty, demonstrating empirically that it generalises effectively across a risk spectrum. CRAC often achieves higher reward than fixed-risk agents, whilst satisfying cost constraints. In cases where CRAC's reward performance is marginally lower than a fixed-risk agent, CRAC retains the advantage of a single risk-conditioned policy that generalises to a risk spectrum, reducing training overhead and providing more control over the reward-cost trade-off.

## 1 Introduction

Constrained Reinforcement Learning (CRL) extends standard Reinforcement Learning (RL) by introducing safety constraints through a dual objective framework; rewards guide task performance whilst costs guide safety objectives (Altman, 1999; Achiam et al., 2017). Policies are trained to maximise rewards whilst ensuring expected accumulated costs remain below predefined safety thresholds. Risk-averse CRL (RaCRL) extends CRL by addressing variability in cost returns arising from environmental uncertainty by controlling for worst-case outcomes using coherent risk measures such as Conditional Value at Risk (CVaR) (Rockafellar & Uryasev, 2000; Yang et al., 2021). However, the likelihood and magnitude of constraint violations caused by uncertainty is generally hard to quantify, leading to difficulty in specifying an appropriate risk level for training an agent at the outset of training.

Current RaCRL approaches adopt a *fixed* level of risk aversion, training policies to satisfy safety constraints at a single, predefined risk level (Yang et al., 2023; Kim et al., 2023; 2024; Yang et al., 2024), effectively committing to a single reward-cost trade-off throughout training and deployment. This prevents the adoption of current RaCRL in problem settings that necessitate flexibility over the reward-safety trade-off. For instance, in autonomous building energy management (Vázquez-Canteli et al., 2019), a user may prefer a highly risk-averse agent under uncertain weather conditions, to mitigate the effect of potential power-outages. Conversely, during favourable weather conditions, the same user may prefer a more risk-neutral agent, that maximises energy savings, through efficient energy use. Fixed risk-averse agents are ill-suited to adapt to these changing preferences, requiring retraining in order to switch between different risk-aversions, limiting their flexibility.

To address this limitation, we propose Conditional Risk-Averse Actor-Critic (CRAC), an off-policy algorithm for RaCRL that learns a single risk-conditioned policy that satisfies risk-based constraints across a spectrum of risk levels. Building on SAC-based RaCRL algorithms, such as Worst-Case Soft Actor-Critic (Yang et al., 2023), CRAC conditions the actor, reward and cost critic's networks directly on a given risk level. CRAC enables exploration and learning across a spectrum of risk levels, allowing the policy network to represent a diverse set of behaviours that balance task and safety objectives in different ways. CRAC's *implicit* exploration is beneficial as fixed-risk policies may insufficiently explore the environment, converging to suboptimal or low reward policies (Greenberg et al., 2022; Yang et al., 2024; McCarthy et al., 2025).

We evaluate CRAC in the Safety-Gymnasium suite of environments (Ji et al., 2023), specifically the Goal1 and Button1 tasks and the Point and Car variants, and in CityLearn (Vázquez-Canteli et al., 2019), a complex building energy management simulation environment. We compare CRAC against two fixed-risk baselines, Worst-Case Soft Actor-Critic (WCSAC) (Yang et al., 2023) and Spectral Risk Constrained Policy Optimisation (SRCPO) (Kim et al., 2024), and a risk-neutral baseline, a SAC-based algorithm Conservative Augmented Lagrangian (CAL) (Wu et al., 2024). Our results demonstrate that CRAC learns constraint satisfying policies across a spectrum of risk levels, often outperforming its fixed-risk counterpart, WCSAC, in terms of reward performance. This demonstrates that risk-conditioning not only provides adaptability to changing risk preferences but can also lead to more favourable trade-offs between safety and task performance. We further demonstrate generalisation to unseen risk levels, and investigate the effect of different risk level sampling during training.

In summary, our main contributions are as follows:

1. We propose CRAC, an off-policy algorithm for RaCRL that learns a single, risk-conditioned policy that adapts to a spectrum of risk-preferences.

2. We empirically demonstrate that CRAC generalises across risk levels and is competitive with or outperforms fixed-risk baselines in terms of the reward-cost trade-off.

## 2 Related Work

**Constrained Reinforcement Learning** Specification of safety in RL is a persistently challenging problem (García & Fernández, 2015; Dulac-Arnold et al., 2021). Constrained Reinforcement Learning (CRL) formulates safety requirements as auxiliary cost functions with corresponding thresholds, separating task objectives from safety objectives (Achiam et al., 2017; Ray et al., 2019; Tessler et al., 2019; Liu et al., 2022; Zhang et al., 2022). These methods typically enforce constraints in expectation through primal-dual optimisation. Off-policy CRL methods, however, suffer from underestimation of cost returns that lead to constraint violations (Zhang et al., 2024; Wu et al., 2024; Gao et al., 2025). To address this, Wu et al. (2024) propose Conservative Augmented Lagrangian (CAL), which over-estimates cost returns to promote conservative behaviour and improved stability in constraint satisfaction. Similarly, stability is improved through PID-based methods (Stooke et al., 2020; Zhang et al., 2024) and the Modified Differential Method of Multipliers (Platt & Barr, 1987; Kumar et al., 2021; Yang et al., 2023). Despite these advances, expectation-based constraints do not explicitly control for rare but catastrophic outcomes.

**Risk-Averse Constrained RL** To mitigate this limitation, risk-averse constrained RL (RaCRL) incorporates risk measures into the constraint. Yang et al. (2021) introduce Worst Case Soft Actor-Critic (WCSAC), approximating worst-case returns of the policy through distributional critics. The authors later extend WC-SAC in (Yang et al., 2023) by incorporating Implicit Quantile Networks (Dabney et al., 2018a), enabling more expressive representations of the cost return distribution. Alternate methods include trust-region approaches with mean-variance style constraints (Kim et al., 2023), on-policy approach using backward value functions for non-stationary policies (Yang et al., 2024), and linearised spectral risk constrained policy optimisation (Kim et al., 2024). Zhang et al. (2025) reformulate the CVaR-constrained problem in an augmented state space and solving the constrained problem through Lagrangian optimisation, deriving near-closed form of the optimal policy. A characteristic shared by these approaches is that a risk level is specified *a priori*, remaining fixed throughout training and deployment.

**Risk- and Constraint-Conditioned RL** In the reward only setting, Choi et al. (2021) and Yoo et al. (2024) propose conditioning agents on a risk parameter, demonstrating improved reward performance over fixed-risk agents, which the authors hypothesise arises from the improved exploration and diversity of experience. In the CRL setting, Yao et al. (2023) propose Constraint-Conditioned Policy Optimisation (CCPO) that trains a CRL policy across various constraint thresholds and adapts to unseen thresholds. In the offline CRL setting Guo et al. (2025) propose Constraint-Conditioned Actor-Critic (CCAC) that conditions the agent's networks on the constraint, regularising cost underestimation in out of distribution (OOD) data through an OOD classifier. Introducing conditional risk-aversion requires approximating the cost return distribution under different policies and evaluating risk-based measures across risk levels. In constraint-conditioned CRL such as CCPO, threshold conditioning changes the allowable expected cost threshold, whereas risk-conditioning changes which part of the cost-return distribution the constraint corresponds to.

Existing RaCRL approaches assume a fixed risk level, and do not learn a single policy capable of generalising across a risk spectrum. Our work addresses this limitation by training a conditional risk-averse policy that jointly optimises reward and risk-based cost constraints across a spectrum of risk levels.

## 3 Background

### 3.1 Risk Constrained RL

A *Constrained Markov Decision Process* (Altman, 1999), is defined by a tuple $(S, A, P, R, C, d)$, where $S$ and $A$ are the state and action spaces respectively, $P : S \times A \times S \to [0, 1]$ is the transition probability function, $R : S \times A \times S \to \mathbb{R}$ is the reward function mapping state-action-next state transitions to a real valued scalar, $C : S \times A \times S \to \mathbb{R}^+$ is the cost function mapping state-action-next state transitions to a non-negative real valued scalar, $d$ is the cost function's corresponding threshold. At each time step $t$, the agent chooses action $a \in A$ given state $s_t \in S$ and policy $\pi(\cdot|s_t)$. The environment responds with a reward $r_t = R(s_t, a_t)$, a cost $c = C(s_t, a_t)$ and next state $s_{t+1} \sim P(\cdot|s_t, a_t)$. We let the random variables $Z_R^\pi(s, a) = \sum_{t=0}^{\infty} \gamma^t r_t$ and $Z_C^\pi(s, a) = \sum_{t=0}^{\infty} \gamma^t c_t$ denote the sum of discounted rewards and costs, respectively, collected along one trajectory of states following $\pi$, beginning in state $s$ and taking action $a$. The risk-averse optimal policy in the CMDP is given by:

$$\max_\pi \mathbb{E}_{s \sim \rho_\pi, a \sim \pi(\cdot|s)}[Z_R^\pi(s, a)] \ \text{ s.t. } \ \mathcal{R}_{s \sim \rho_\pi, a \sim \pi(\cdot|s)}^\sigma[Z_C^\pi(s, a)] \leq d \tag{1}$$

where $\rho_\pi$ denotes the steady state distribution of $\pi$ and $\mathcal{R}^\sigma$ is spectral risk measure (Acerbi, 2002; Kim et al., 2024). Throughout this paper we use Conditional Value at Risk (CVaR) (Rockafellar & Uryasev, 2000), as our spectral risk measure, quantifying the expected value of the worst $\alpha$ percentile of cost returns under $\pi$:

$$\text{CVaR}_\alpha[Z_C^\pi] = \mathrm{E}[z_c|Z_C^\pi \geq F_{Z_C^\pi}^{-1}(1 - \alpha)] \tag{2}$$

where $F_{Z_C^\pi}^{-1}$ is the quantile function of the random variable $Z_C^\pi$, and $\alpha \in [0, 1]$ is our desired risk level. Yang et al. (2023) propose Worst-Case Soft Actor-Critic that aims to solve this risk-averse CMDP problem through the following primal-dual optimisation objective:

$$\min_{\lambda \geq 0} \max_\pi \mathbb{E}_{s \sim \rho_\pi, a \sim \pi(\cdot|s)}[Q_R^\pi(s, a) - \lambda(Q_{C,\alpha}^\pi(s, a) - d)] \tag{3}$$

where $Q_R^\pi(s,a)$ is the reward value function approximation of $\mathbb{E}[Z_R^\pi(s,a)]$, and $Q_{C,\alpha}^\pi(s,a)$ is the cost value function approximation of the $\text{CVaR}_\alpha[Z_C^\pi(s,a)]$. Next, we discuss learning a parameterised distributional critic to approximate $Q_{C,\alpha}^\pi(s,a)$

## 3.2 Distributional RL

The full quantile function of the state-action return distribution, can be approximated through Distributional RL and quantile regression (Dabney et al., 2018b). Letting $F_{Z_C^\pi}^{-1}$ denote the quantile function of the random variable $Z_C^\pi$ we learn a parametrised quantile function $Z^\pi(s,a|\theta_C)$, mapping the quantile fraction $\tau \in [0,1]$ to the quantile value $F_{Z^\pi}^{-1}(\tau)$. The parametrised quantile function $Z^\pi(s,a|\theta_C)$ is trained by minimising the quantile regression loss (Dabney et al., 2018b; Yang et al., 2023; Yoo & Woo, 2025):

$$\mathcal{L}(\theta_C) = \frac{1}{NN'}\sum_{i=0}^{N}\sum_{j=0}^{N'}|\tau_i - 1(\delta_{ij} < 0)| \cdot |\delta_{ij}| \tag{4}$$

where $\delta_{ij}$ is the pairwise temporal difference error between target distribution $\bar{Z}(s,a|\tau_j)$ and current distribution $Z(s,a|\tau_i)$ defined:

$$\delta_{ij} = c_t + \gamma \bar{Z}_C^\pi(s_{t+1},a_{t+1}|\tau_j,\bar{\theta}_C) - Z_C^\pi(s_t,a_t|\tau_i,\theta_C) \tag{5}$$

where $\tau_i, \tau_j$ denote $N, N'$ total independent random quantiles drawn from a uniform distribution $U[0,1]$. Then using this parametrised quantile distribution we can approximate $\text{CVaR}_\alpha(Z_C^\pi)$:

$$Q_{C,\alpha}^\pi(s,a) = \frac{1}{\alpha}\int_{1-\alpha}^{1} Z_C^\pi(s,a|\theta_C)(\tau)d\tau \tag{6}$$

## 4 Conditional Risk-Averse Actor Critic

In this section, we present our problem definition and outline the proposed algorithm.

### 4.1 Problem Specification

Our aim is to learn a risk-conditioned policy $\pi(a|s;\alpha)$ that maps states $s \in S$ and risk levels $\alpha \in \mathcal{A} \subseteq (0,1]$, to a distribution over actions $a \in A$, such that the resulting policy satisfies the risk measure placed on the cost function $C$ for all risk levels in $\mathcal{A}$. Formally, our objective is:

$$\max_{\pi_\alpha} \mathbb{E}_{s\sim\rho_{\pi_\alpha},a\sim\pi(\cdot|s,\alpha)}[Z_R^{\pi_\alpha}(s,a)] \ \ \textbf{s.t.} \ \mathcal{R}_{s\sim\rho_{\pi_\alpha},a\sim\pi(\cdot|s,\alpha)}^{\sigma_\alpha}[Z_C^{\pi_\alpha}(s,a)] \le d; \ \forall \alpha \in \mathcal{A} \tag{7}$$

where $\mathbb{E}[Z_R^{\pi_\alpha}(s,a)]$ is the expected reward return following the risk-conditioned policy $\pi_\alpha$, $\mathcal{R}_{\sigma_\alpha}$ is a spectral risk-measure (e.g. CVaR) at a given risk level $\alpha$ calculated over $Z_C^{\pi_\alpha}(s,a)$ the distribution of cost returns collected following the risk-conditioned policy $\pi_\alpha$, and $d$ is the cost constraint, fixed across all risk levels.

### 4.2 Algorithm

To learn a policy that satisfies our objective in Eq. 7, we propose Conditional Risk-Averse Actor-Critic (CRAC), an off-policy algorithm for RaCRL, extending the fixed-risk algorithm Worst-Case Soft Actor-Critic (WCSAC) (Yang et al., 2023), to a *continuous spectrum* of risk levels. Specifically, we condition the actor, the reward and cost critic networks on a given risk parameter $\alpha$, and train a risk-conditioned Lagrangian multiplier, allowing a single set of networks to represent policies across a spectrum of risk levels. Algorithm 1 summarises the full procedure, and below we discuss in more detail the key extensions made to the fixed-risk algorithm.

During exploration, CRAC samples a risk level $\alpha_e$ at the beginning of each episode, and fixes it for the entire episode (lines 3-9). During learning, CRAC samples mini-batches of replay experience, $(s, a, s', r, c)$,

---

**Algorithm 1** Conditional Risk-Averse Actor-Critic (CRAC)

---

1: **Initialize:** Policy $\pi_\phi(a|s; \alpha)$, reward-critics $Q_{\theta_R}(s, a; \alpha)$, distributional cost-critics $Z_{\theta_C}(s, a; \alpha)$, target critics $\{\bar{Q}_{\theta_R}, \bar{Z}_{\theta_C}\}$, Lagrange multiplier network $\lambda(\alpha)$, risk cut-offs $[\alpha_{\min}, \alpha_{\max}]$, replay buffer $\mathcal{B}$
2: **for** each iteration **do**
3:     Sample exploration risk level $\alpha_e \sim U[\alpha_{\min}, \alpha_{\max}]$             ▷ Sample $\alpha_e$ at beginning of episode
4:     **for** each environment step **do**
5:         Select action $a_t \sim \pi_\phi(\cdot|s_t; \alpha_e)$                     ▷ Condition actor on $\alpha_e$
6:         Execute $a_t$, obtain $s_{t+1}, r_t, c_t$
7:         Store $(s_t, a_t, r_t, c_t, s_{t+1})$ in $\mathcal{B}$
8:     **end for**
9:     **for** each gradient step **do**
10:         Sample N transitions $\{(s, a, r, c, s')\} \sim \mathcal{B}$
11:         Sample training risk levels $\alpha_{\mathrm{tr}} \sim U[\alpha_{\min}, \alpha_{\max}]$
12:         Update Conditional Reward Critic                     ▷ Eq. 9
13:         Update Conditional Cost Critic                      ▷ Eq. 4
14:         Update Conditional Actor                          ▷ Eq. 13
15:         Update Conditional Lagrangian Multiplier          ▷ Alg. 2
16:     **end for**
17: **end for**

---

from the replay buffer, sampling a new risk level $\alpha_{tr}$ from a uniform distribution $\alpha_{tr} \sim U[\alpha_{\min}, \alpha_{\max}]$, where $\alpha_{\min}, \alpha_{\max}$ define the range of the spectrum, with smaller values of $\alpha$ indicating more risk-averse policies (line 12). This experience is then used to update the conditional actor, reward and cost critics and the conditional Lagrangian multiplier.

**Conditional Reward Critic** We extend the reward critic to approximate the return of the risk-conditioned policy $\pi_\alpha$ by conditioning on the reward critic on $\alpha$:

$$Q_R^{\pi_\alpha}(s, a|\alpha) = \mathbb{E}[Z_R^{\pi_\alpha}(s, a)] \tag{8}$$

We train this reward critic by minimising the mean squared TD-error:

$$L(\theta_R) = \mathbb{E}_{(s,a,r,s',\alpha)}\left[\left(Q_R^{\pi_\alpha}(s, a|\alpha) - y_R\right)^2\right] \tag{9}$$

where $y_R$ is the SAC Bellman target:

$$y_R = r_t + \gamma \mathbb{E}_{s_{t+1} \sim B, a_{t+1} \sim \pi(\cdot|s_{t+1}, \alpha)}\left[\bar{Q}_R(s_{t+1}, a_{t+1}|\alpha) - \beta \log \pi(a_{t+1}|s_{t+1}, \alpha)\right] \tag{10}$$

**Conditional Cost Critic** To estimate the CVaR cost returns of policy $\pi_\alpha$ we approximate the full cost return distribution through the quantile critic $Z_C^{\pi_\alpha}(s, a|\tau, \alpha)$ that approximates the quantile function $F_{Z^{\pi_\alpha}}^{-1}(\tau)$ for policy $\pi_\alpha$. The Bellman target distribution is then:

$$Z_C = c_t + \gamma \bar{Z}_C^{\pi_\alpha}(s_{t+1}, a_{t+1}|\tau_j, \alpha) \tag{11}$$

where $a_{t+1} \sim \pi(\cdot|s_{t+1}, \alpha)$ and $\tau_j \sim U[0, 1]$. The cost critic parameters are updated using the quantile regression loss, replacing the pairwise temporal difference error in Eq. 5 with risk-conditioned value estimates:

$$\delta_{ij} = Z_C - Z_C^{\pi_\alpha}(s_t, a_t|\tau_i, \alpha) \tag{12}$$

where $\tau_i \sim U[0, 1]$, sampled separately from $\tau_j$. CVaR cost returns of policy $\pi_\alpha$ are approximated through:

$$Q_{C,\alpha}^{\pi_\alpha}(s, a|\alpha) = \frac{1}{\alpha} \int_{1-\alpha}^1 Z_C^{\pi_\alpha}(s, a|\alpha)(\tau) d\tau \tag{13}$$

where in practice we approximate this by averaging over a distorted quantile sampling distribution $\tau_k \sim U[(1 - \alpha), 1]$, similarly to Dabney et al. (2018a); Yang et al. (2023).

---

**Algorithm 2** Risk-Conditioned PID Lagrange Multiplier

---

1: **Initialise:** Networks $f_I(\alpha)$ (integral) and $f_{prev}(\alpha)$ (previous-cost), Gains $K_P, K_I, K_D \geq 0$, cost limit $d$
2: **repeat** at each iteration:
3:     Receive risk level $\alpha$ and cost estimate $J_C(\alpha)$
4:     Predict states: $I(\alpha) \leftarrow f_I(\alpha), \quad J_{C,\text{prev}}(\alpha) \leftarrow f_{\text{prev}}(\alpha)$
5:     Constraint error: $\Delta(\alpha) \leftarrow \big(J_C(\alpha) - d\big)$
6:     Derivative proxy: $\partial(\alpha) \leftarrow \big(J_C(\alpha) - J_{C,\text{prev}}(\alpha)\big)_+$
7:     Integral update: $I(\alpha) \leftarrow \big(I(\alpha) + \Delta(\alpha)\big)_+$
8:     Update networks:
9:         $I^{\text{tgt}}(\alpha) \leftarrow I(\alpha)$
10:         $J_{C,\text{prev}}^{\text{tgt}}(\alpha) \leftarrow J_C(\alpha)$
11:         Update $f_I, f_{\text{prev}}$ by minimising $\mathcal{L}_{PID} = \text{MSE}(I, I^{\text{tgt}}) + \text{MSE}(J_{C,\text{prev}}, J_{C,\text{prev}}^{\text{tgt}})$
12: **return** Multiplier: $\lambda(\alpha) \leftarrow \big(K_P \Delta(\alpha) + K_I I(\alpha) + K_D \partial(\alpha)\big)_+$

---

**Conditional Actor**     The risk-conditioned actor is trained to maximise the risk-conditioned objective:

$$J(\pi) = \text{E}_{s \sim \mathcal{B}, a \sim \pi(\cdot|s,\alpha), \alpha \sim \rho}\left[Q_R^{\pi_\alpha}(s,a|\alpha) - \lambda(\alpha)\big(Q_{C,\alpha}^{\pi_\alpha}(s,a|\alpha) - d\big) - \beta \log \pi(a|s,\alpha)\right] \tag{14}$$

where $\rho$ is the risk level sampling distribution, $Q_R^{\pi_\alpha}(s,a|\alpha)$ is the state-action reward value of the policy $\pi_\alpha$, $Q_{C,\alpha}^{\pi_\alpha}(s,a|\alpha)$ is the risk-based state-action cost value at the risk level $\alpha$ of the policy $\pi_\alpha$, $\lambda(\alpha)$ is the Lagrangian multiplier conditioned at risk level $\alpha$, and $\beta$ is a tunable parameter that controls the actor's entropy (Haarnoja et al., 2019). To represent the Lagrangian multiplier across the spectrum of risk levels, we make use of a neural network based representation of a PID Lagrangian Controller (Stooke et al., 2020).

**Conditional Lagrangian Multiplier**     We parametrise the Lagrangian multiplier as a neural network $\lambda(\alpha) \geq 0$, projecting its output to ensure non-negative values. Since PID control has been shown to improve the stability of Lagrangian based CRL by reducing overshoot and oscillations (Stooke et al., 2020; Ray et al., 2019; Ji et al., 2023; Zhang et al., 2024), we extend the PID Lagrangian approach of (Stooke et al., 2020) to a conditional Lagrangian multiplier. Specifically, $\lambda(\alpha)$ takes as input a risk level $\alpha$, outputting the PID Lagrangian multiplier, as summarised in Algorithm 2. An MLP approximates the risk-conditioned Integral term (line 4), corresponding to a standard Lagrangian multiplier network. We then add the Proportional term that immediately responds to constraint violations, and a Derivative term using a second separate MLP to learn a risk-conditioned error term. Each MLP approximates a running estimate of the corresponding PID state by minimising mean squared error loss between predicted and target values (line 9-11).

## 5   Experiments and Results

In this section, we present our three-fold evaluation of CRAC. First, we evaluate CRAC against two fixed risk-averse baseline algorithms, and a risk-neutral baseline, presenting this comparison in Section 5.2.1. Second, in Section 5.2.2, we investigate the generalisation of CRAC to unseen risk levels, by limiting the spectrum of risk levels it is trained on and evaluating it outside of this spectrum. Finally, we present an ablation study of key ingredients of CRAC in Section 5.3, namely we evaluate its performance under three distinct risk level sampling schemes and compare its performance using a traditional conditional Lagrangian multiplier vs a PID style Lagrangian multiplier.

### 5.1   Experimental Setup

#### 5.1.1   Environments

The Safety-Gymnasium environments, presented by Ji et al. (2023), act as the standard baseline environments for constrained RL. Reward performance generally decreases with added risk-aversion as agents

become increasingly conservative in exploring the environment. In Goal1, the objective is to reach a goal position whilst avoiding static obstacles. In Button1, the objective is to press one of a set of randomly positioned buttons, whilst avoiding both static and moving obstacles, making it the more challenging environment variant. We carry out experiments in both the Point and Car variants of the Goal1 and Button1 environments, with the Car variant being the more challenging of the two as the agent controls a larger Car object, making navigation more challenging. Each environment randomly generates a layout of obstacles at the beginning of each episode and the goal changes position randomly each time the agent reaches it during an episode. The observation spaces consist of a large number of features representing lidar readings of the agent, describing the distance to obstacles and the goal or button. The reward function is a dense function measuring the absolute distance towards the goal and a constant for reaching the goal location or pushing the correct button. Unsafe behaviour of colliding with an obstacle, entering a static obstacle zone or pushing the wrong button returns a cost of 1, whilst safe behaviour returns a cost of 0. We set the cost constraint to 10 for each episode and truncate the episodes to a max length of 400 steps, adapting changes described in (Liu et al., 2022; Wu et al., 2024) to speed up training of the agents, by increasing the simulation timesteps of the environment by 2.5-3x, but otherwise the environments remain the same.

CityLearn (Vázquez-Canteli et al., 2019), a real-world simulation environment, presents a challenging environment where the agent is tasked with minimising the electricity demand the building places on the grid, whilst maintaining the thermal comfort of a building, by ensuring temperature of the building be within 1 degree of the desired set-point temperature. Power-outages occur in the environment randomly and infrequently, cutting the supply of electricity from the grid to each building, and can last for a number of hours. During regular operations the agent uses energy supplied from the grid to control the temperature of the building but during outages it must rely on either solar energy or energy stored in its electrical battery. A risk-averse policy should learn to prepare for power-outages by storing electricity in the battery. We set the cost constraint to 720 and each episode has a length of 720 timesteps.

### 5.1.2 Algorithms

Our aim is to compare the reward-cost trade-off of fixed risk-aversion and conditional risk-aversion, across a spectrum of risk-aversion levels. First, we establish the risk-neutral reward performance in each environment using the Conservative Augmented Lagrangian (CAL) algorithm (Wu et al., 2024), a risk-neutral state-of-the-art off-policy Soft Actor-Critic method. We then use two fixed risk-averse baselines, an off-policy algorithm Worst-Case Soft Actor-Critic (WCSAC) (Yang et al., 2023), and a recent state-of-the-art baseline Spectral Risk Constrained Policy Optimisation (SRCPO) (Kim et al., 2024). We compare each of these algorithms against our proposed algorithm CRAC, measuring in particular the constraint satisfaction and their reward-cost trade-off against CAL, with increasing risk-aversion. We train each algorithm across 5 seeds, for 5 million timesteps in Safety-Gymnasium and 3 million timesteps in CityLearn. We train each fixed risk-averse algorithm under the following risk levels [0.1, 0.25, 0.5, 0.75, 1.0], with 0.1 being the most risk-averse and 1.0 being risk-neutral. We train CRAC across a continuous spectrum of risk levels $\alpha \in [0.05, 1]$.

We reimplement CAL[1], and use the official codebase for SRCPO[2], using the hyperparameters reported in the original implementations. While both choices follow reported hyperparameter setups, performance of each may vary in our experimental setup due to different factors such as hardware, random seeds and our implementation details. We therefore interpret the results for CAL and SRCPO as strong baseline implementations under the reported hyperparameters, rather than exhaustively tuned baselines. We discuss this further in the supplemental material.

### 5.1.3 Implementation Details

We describe two implementation choices applied to both CRAC and its fixed-risk counterpart WCSAC. These are not core contributions of this work but affect stability and performance of CRAC and WCSAC [3].

---

[1] https://github.com/ZifanWu/CAL
[2] https://github.com/rllab-snu/Spectral-Risk-Constrained-RL
[3] We include ablations of each algorithm's performance with and without these components in the supplemental material.

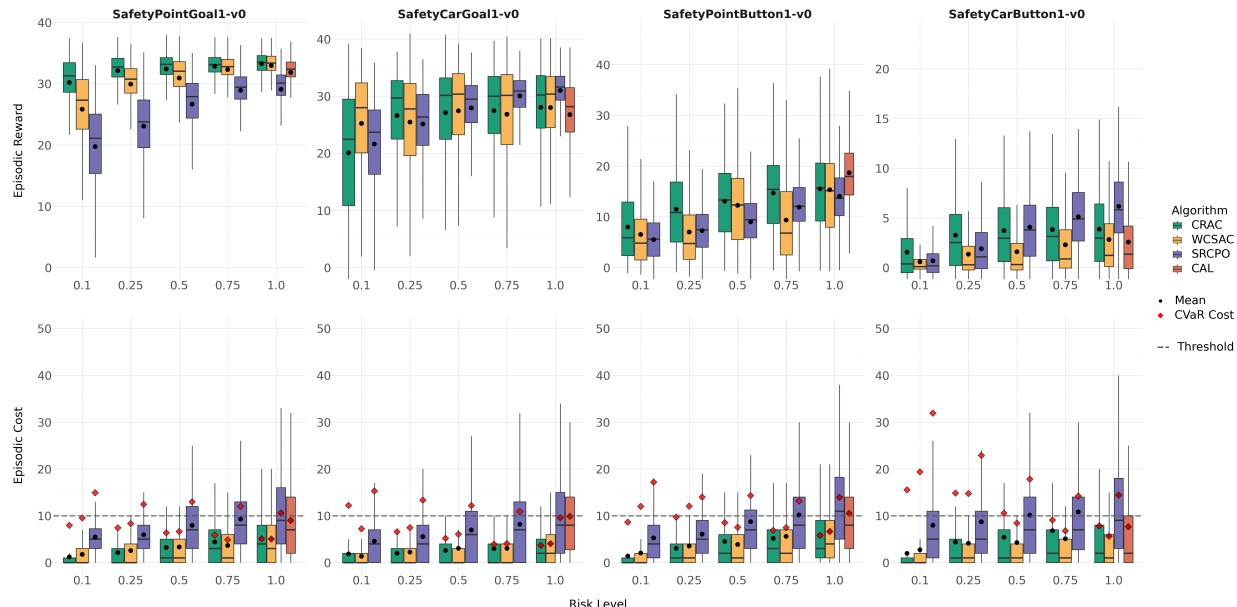

Figure 1: **Safety-Gymnasium Reward-Cost Distributions.** Box plots of evaluation results for fixed vs conditional risk comparison. Each algorithm is trained with 5 random seeds and evaluated for 100 episodes. Each fixed risk-averse algorithm is trained separately at the fixed risk levels shown. For clarity, outliers are removed, with red diamonds indicating CVaR Cost for each algorithm.

**Cost Augmented State** Risk-constrained RL problems are non-Markovian and necessitate a history conditioned policy (Kim et al., 2024; Zhang et al., 2025). Work by Bastani et al. (2022) demonstrate that an optimal history conditioned policy can be defined in an augmented state space, which incorporates the discounted sum of costs. As such, we augment the state space with the discounted sum of costs as follows [4]:

$$\bar{s}_t = (s_t, e_t, b_t), \text{ where } e_0 = 0, \ e_{t+1} = (e_t + C(s_t, a_t, s_{t+1}))/\gamma, \ b_t = \gamma^t \tag{15}$$

For consistency of evaluation, we apply this to each algorithm.

**Truncated Quantile Critics** CRAC extends fixed-risk SAC based algorithms to a continuous spectrum of risk levels. As such, it inherits known challenges of off-policy CRL, namely cost underestimation due to bootstrapping bias in cost critics (Wu et al., 2024; Zhang et al., 2024). To address this we apply truncation to the distributional cost critic following Truncated Quantile Critics (Kuznetsov et al., 2020). Given N cost critics each predicting M quantiles of the policy's cost return distribution $Z_C^{\pi_\alpha}(s, a|\alpha)$, TQC formulates the target in the TD-learning step as a sorted mixture of $N \times M$ quantiles. In the original form of TQC, $K$ quantiles are truncated from the upper tail of the distribution to mitigate over-estimation of reward, however, as we are concerned with the upper tails of the cost distribution in RaCRL, we truncate $K$ quantiles from the lower tail. For our WCSAC and CRAC we use Implicit Quantile Networks (Dabney et al., 2018a) for the cost critics, applying truncation in Safety-Gymnasium only, due to the sparsity in cost.

### 5.2 Results

#### 5.2.1 Conditional Risk vs Fixed Risk

**Safety-Gymnasium** In Figure 1 we present the results of our comparative experiments in the Safety-Gymnasium environments, where each fixed-risk algorithm is trained at a number of risk levels. We observe that CRAC shows more favourable reward performance when compared against the fixed risk-averse algorithms, except in the CarGoal1 environment. In the Point environments, CRAC achieves the best results

---

[4]In practice, $e_t$ can grow to extremely large values when $\gamma < 1$, especially in long horizon problems, and destabilise the learning of function approximators, we therefore use $\log(e_t + 1)$. We include ablations of each algorithm's performance with and without this augmented state in the supplemental material.

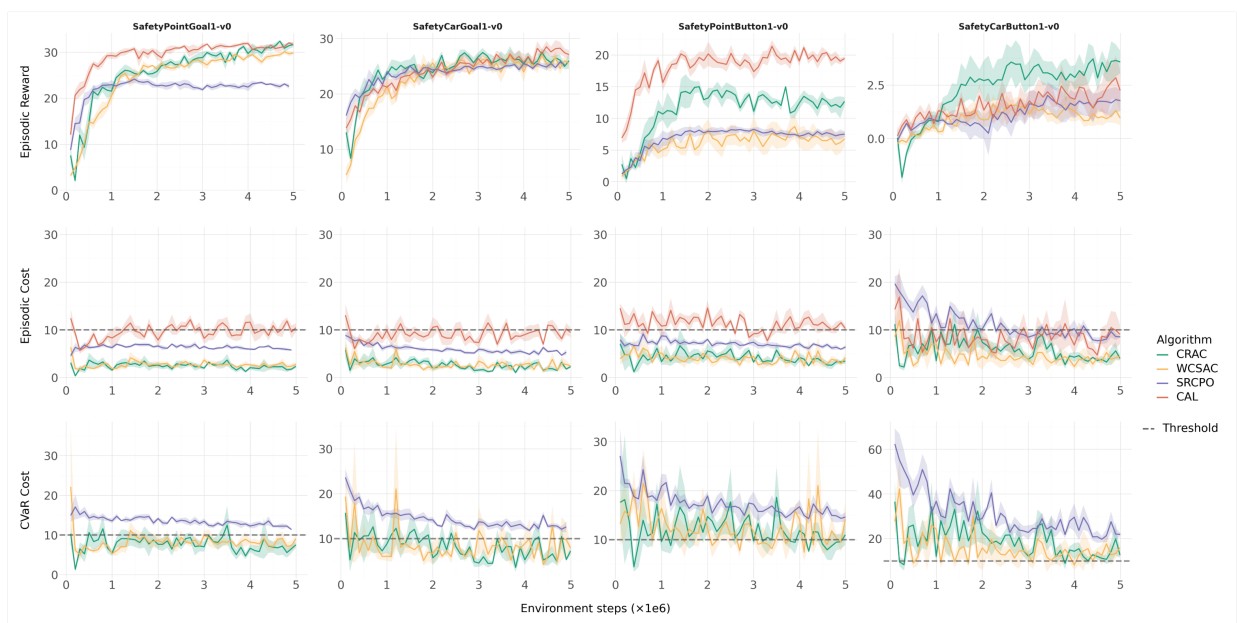

Figure 2: **Safety-Gymnasium training curves.** WCSAC and SRCPO are trained at risk level $\alpha = 0.25$; CRAC is trained on a full risk spectrum $[0.05, 1]$ and evaluated at $\alpha = 0.25$; CAL is trained to satisfy expected constraints. Solid lines show the mean; shaded regions show the standard deviation across 5 seeds (scaled by 0.2). For clarity we exclude CAL from "CVaR Cost" plot as it is trained to satisfy expected cost constraints.

by satisfying the CVaR based constraints whilst achieving the highest reward. In CarGoal1 we observe that CRAC violates the CVaR based constraints at the most risk-averse level $\alpha = 0.1$, whilst WCSAC satisfies it and achieves higher reward. At other risk levels it is competitive in reward with WCSAC, sometimes outperforming it, whilst satisfying the CVaR constraints. In CarButton1, we note that all algorithms fail to satisfy the CVaR based constraints at the lower risk levels $\alpha \in [0.1, 0.25]$, whilst exhibiting a low distribution of costs. This indicates a common challenge in the Safety-Gymnasium environments of rare but high cost episodes (Ji et al., 2023). However, as satisfying CVaR constraints becomes increasingly more difficult as $\alpha$ tends towards more risk-averse values, constraint satisfaction may become near-infeasible at more risk-averse levels, under a fixed constraint. We further note that SRCPO fails to converge to solutions that satisfy the CVaR based constraints in all environments [5].

In summary, our results demonstrate that CRAC improves the reward-cost trade-off of fixed-risk agents, in particular improving the trade-off over WCSAC, its fixed-risk variant. This indicates that the implicit exploration of CRAC under various risk levels improves the reward performance across the spectrum of risk levels, particularly at lower risk levels where the conservative nature of risk-averse exploration hampers its exploration and learning (McCarthy et al., 2025). This is further demonstrated in Figure 2, showing that CRAC achieves higher reward throughout training when compared against both WCSAC and SRCPO, reducing the loss in reward performance against the risk-neutral baselines CAL.

**CityLearn** We conduct comparative experiments in the CityLearn environment (Vázquez-Canteli et al., 2019). Results are presented in Figure 3. We observe that the fixed-risk baseline, WCSAC, achieves marginally better reward-cost trade-off for constraint-satisfying solutions at risk levels $\alpha \in [0.1, 0.25, 0.5]$ compared to CRAC [6].

---

[5]In their original work, Kim et al. (2024) trained SRCPO under a CVaR alpha of 0.25 and presented the expected cost rate that satisfies the cost thresholds, rather than presenting the CVaR cost rates.

[6]We were unable to train a SRCPO policy to converge to a meaningful solution, so have therefore excluded it

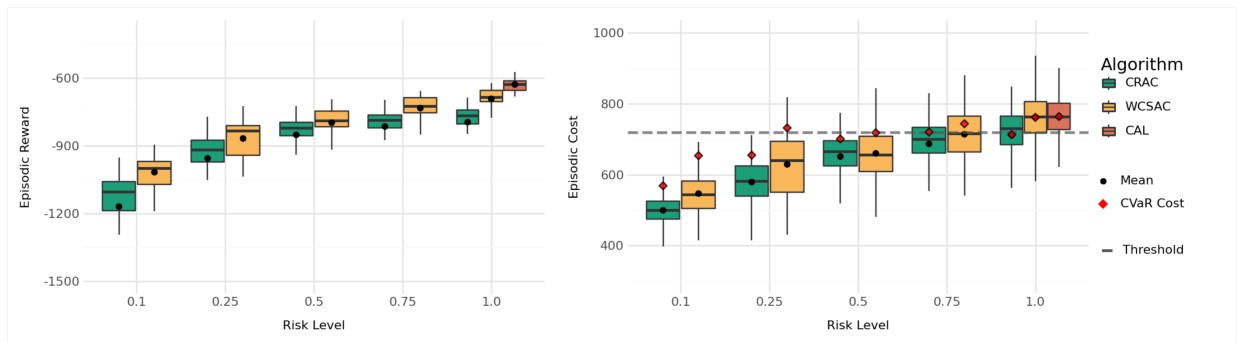

Figure 3: **CityLearn Reward-Cost Distributions.** Box plots of evaluation results for fixed vs conditional risk comparison. Each algorithm is trained with 5 random seeds and evaluated for 100 episodes. Each fixed risk-averse algorithm is trained separately at the fixed risk levels shown. For clarity, outliers are removed, with red diamonds indicating CVaR Cost for each algorithm.

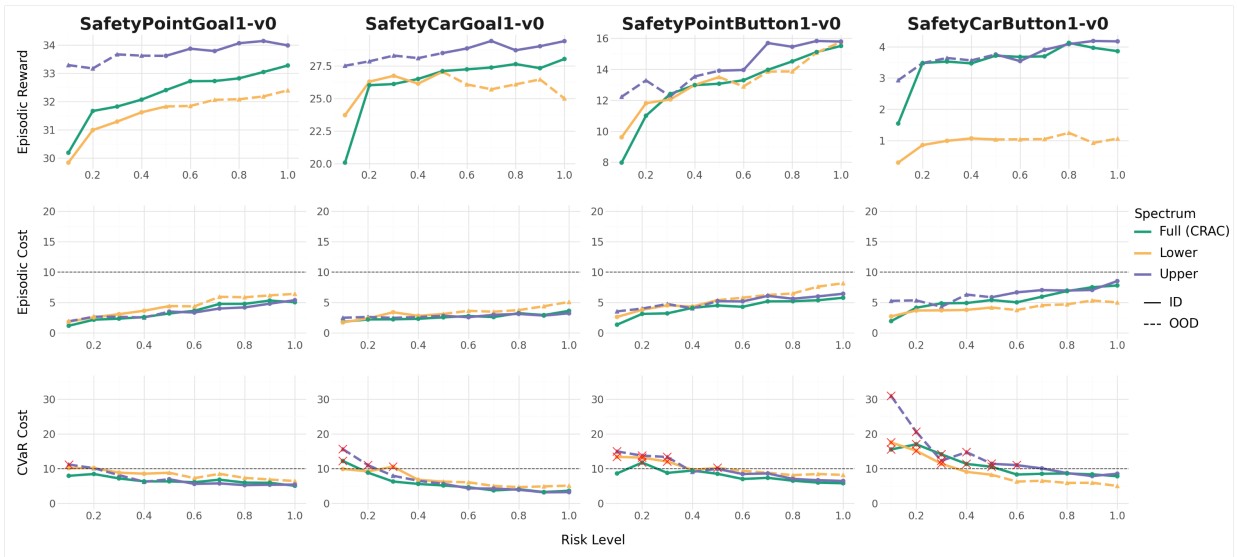

Figure 4: **Safety-Gymnasium generalisation to unseen risk levels.** (i): *Lower* - Risk-averse biased spectrum $\alpha \in [0.05, 0.5]$, (ii): *Upper* - Risk-neutral biased spectrum $\alpha \in [0.5, 1.0]$. (iii): *Full* - complete risk spectrum $\alpha \in [0.05, 1.0]$. *ID* indicates evaluation at risk levels on which the agent was trained, *OOD* indicates evaluation at risk levels outside of the agent's training spectrum.

This comparison, however, highlights a key advantage of the conditional risk-aversion achieved by CRAC, over fixed risk-aversion. A single CRAC agent is trained to satisfy the constraint across the full spectrum of risk levels, whilst achieving competitive reward performance relative to its fixed-risk counterpart. In contrast, WCSAC requires training separate agents for each risk level. This highlights the primary practical benefit of CRAC, it provides a single, risk-conditioned policy capable of representing the full reward-cost trade-off across a spectrum of risk levels. As such, CRAC is practically valuable to end users who require tuneable risk-aversion or may wish to explore how risk-aversion impacts the reward-cost trade-off, without incurring the computational overhead of training multiple risk-averse agents.

### 5.2.2 Generalisation to Unseen Risk Levels

To evaluate CRAC's ability to generalise to risk levels beyond its training distribution, we train two sets of CRAC agents under truncated risk level spectra: (i) a risk-neutral biased spectrum, $\alpha \in [0.5, 1]$, and (ii)

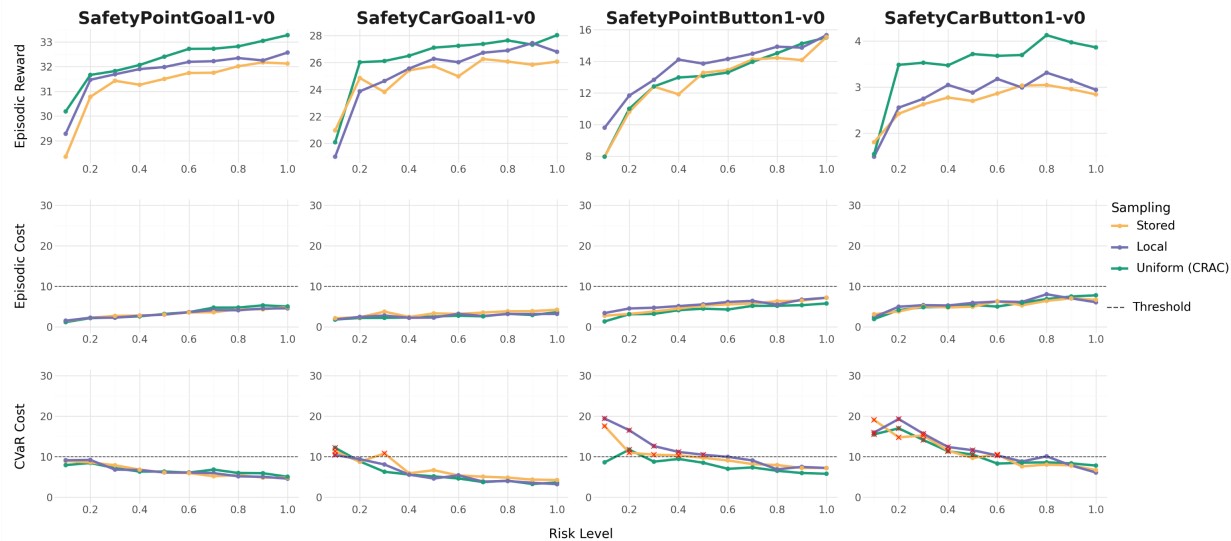

Figure 5: **Ablation of three risk level resampling variants.** (i): *Stored* - No resampling, (ii): *Local* - Local uniform resampling centred on stored risk level, (iii): *Uniform* - Full uniform sampling agnostic to stored risk level

a risk-averse biased spectrum, $\alpha \in [0.05, 0.5]$. Each agent is then evaluated on risk levels outside of their respective training risk spectra, with results shown in Figure 4.

When trained on the risk-neutral biased spectrum, CRAC achieves higher levels of reward than the full spectrum agent. However, its cost performance degrades as evaluation shifts towards more risk-averse levels. In particular, it fails to satisfy CVaR-based constraints at the risk-averse levels furthest from its training data. For the risk-averse biased spectrum, $\alpha \in [0.05, 0.5]$, we observe two trends. Firstly in the Goal environments, the policy converges to a constraint satisfying policy, but achieves lower reward than the full-spectrum CRAC in the PointGoal environment and slightly better reward in CarGoal. In PointButton, the risk-averse biased policy converges to a higher reward policy at more strongly risk-averse levels $\alpha \in [0.1, 0.3]$, when compared to the full-spectrum CRAC. It fails, however, to satisfy the CVaR based constraints at these levels, whilst the full spectrum CRAC satisfies them. In the CarButton1 environment, the risk-averse biased agent achieves the lowest reward but generally the lowest cost.

Together, these results highlight an important limitation of CRAC's generalisation across risk levels, OOD generalisation appears to be asymmetric. When the risk-neutral biased agent is evaluated at more risk-averse levels outside of its training data, generalisation degrades. While training under more risk-neutral risk levels leads to less conservative exploration and higher reward, without explicit exposure to more risk-averse levels, constraint satisfaction does not reliably transfer further from its training data. Conversely, training on more risk-averse levels, leads to lower reward policies but better generalisation towards risk-neutral levels outside of its training data. Lastly, training across the full spectrum shows signs of a regularising effect, in all environments bar the CarButton environment, by leading to lower cost than the risk-averse biased agent.

### 5.3 Ablations

In this section, we report the results of an ablation study of the two key components of CRAC: (i) risk level sampling during learning and (ii) the formulation of the conditional Lagrangian multiplier.

#### 5.3.1 Risk Level Sampling

In Figure 5 we present comparative results of three risk level sampling approaches in the Safety-Gymnasium environments: (i) *Stored* represents CRAC trained using the risk level sampled during the exploration step

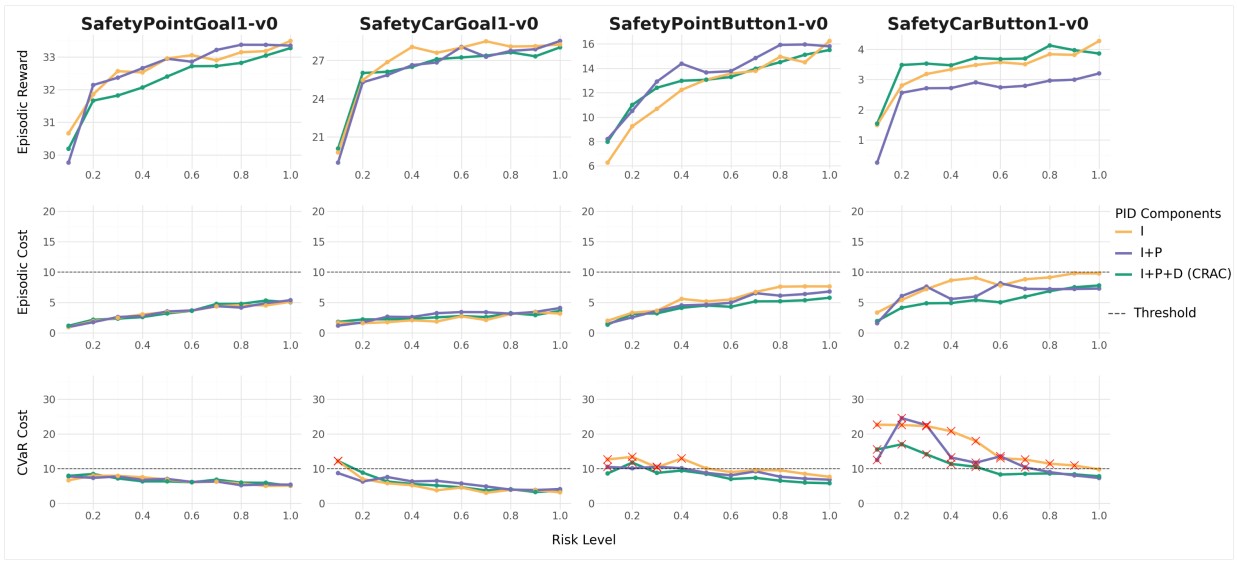

Figure 6: **Ablation of PID components.** (i): $I$ - Integral only term responding to steady-state violations (closest to standard Lagrangian), (ii): $I + P$ - Integral and Proportional Term responding to each sample's cost violations, (iii): $I + P + D$ - Integral, Proportional and Derivative terms combined in full PID approach. Derivative term responds to changes in cost

only, (ii) *Local* represents CRAC trained by resampling the stored risk level from a uniform distribution centred at the stored level, where at boundaries we ensure a minimum sampling width of 0.1 around the stored level, and (iii) *Uniform* represents full uniform risk level resampling, agnostic to the stored risk level.

We observe a clear improvement in CRAC's reward in all environments under both the *Local* and the *Uniform* sampling approaches, except in PointButton1 where each approach achieves similar reward. This indicates that relabelling transition samples significantly improves CRAC's ability to learn higher reward behaviour, by improving the diversity of replay experience available to CRAC through implicit exploration.

We further observe that in all environments, *Uniform* sampling largely improves the cost satisfaction of CRAC over the other two sampling approaches. In particular, at more risk-averse risk levels, learning under *Stored* and *Local* sampling approaches leads to cost violations, indicating that *Uniform* sampling leads to more conservative behaviour, whilst having little to no impact on reward performance. We hypothesise that for more risk-averse risk levels, *Uniform* sampling has similar policy regularisation effects to cost value overestimation techniques such as that of CAL (Wu et al., 2024), by evaluating the cost values at states collected by more risky policies. States may have higher cost values when collected under a risk-neutral biased policy and used to train a more risk-averse biased policy, leading to higher estimated cost returns.

### 5.3.2 Conditional Lagrangian Multiplier

We analyse the contribution of each PID component to the final performance of CRAC by training three variants of the conditional Lagrangian multiplier: (i) Integral only (I) – represents a simple conditional Lagrangian multiplier most similar to standard approaches, responding to steady-state constraint violations, (ii) Integral and Proportional (I+P) – we add the proportional term that responds immediately to constraint violations, and (iii) Integral, Proportional and Derivative (I+P+D) – we use the complete PID formulation with the derivative term to respond to changes in previous errors. For these experiments, and all experiments, we set the value of PID hyperparameters to I=0.0005, P=0.1 and D=0.0005.

In Figure 6 we observe that CRAC is quite robust to the combination of PID components used in the Goal1 environments, satisfying cost constraints and showing similar reward scores. However, in the Button1 environments, we observe a clear performance improvement in cost when the full PID setup is used, with the

largest improvement in CarButton1. These results indicate that the performance improvements observed in CRAC cannot be attributed solely to risk-conditioning, but also partially to the stability introduced by incorporating the PID Lagrangian update, particularly in the more challenging Button environments. To better disentangle this, we have included a comparison between CRAC and WCSAC with PID in Appendix Section 11, isolating the contribution of the dual update mechanism from that of risk-conditioning.

## 6    Conclusion

In this work we presented CRAC, a conditional risk-averse actor-critic algorithm for constrained RL that trains a single risk-conditioned policy across a continuous spectrum of risk. Unlike fixed-risk algorithms that require separate training for a single risk level, CRAC trains one policy that generalises across a continuous risk spectrum. CRAC enables tunable deployment of risk-based constrained RL agents based on a balance between reward-cost trade-off and risk preferences. Across multiple environments we showed that CRAC achieves competitive performance against fixed-risk baselines, often outperforming them, indicating that learning across multiple risk levels helps mitigate the conservatism introduced by risk-aversion, while the PID Lagrangian update improves training stability. Whilst we focused on static risk level sampling schemes, future work may look at how risk level sampling can improve the sample efficiency of CRAC by dynamically responding to the performance of CRAC at specific risk levels.

### Acknowledgments

The author would like to acknowledge the support of the IBM PreDoc Programme, part-funded by IDA Training Grant 215544

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

## A    Hyperparameters

In this section we outline the hyperparameters selected for each algorithm. We reimplement CAL, making use of the official hyperparameters used for CAL (https://github.com/ZifanWu/CAL), rather than performing expensive hyperparameter tuning, as the authors' original paper presented a number of ablations over the key parameters introduced by CAL (Wu et al., 2024). We use the official code repository for SRCPO (https://github.com/rllab-snu/Spectral-Risk-Constrained-RL), making use of their official hyperparameters also, rather than performing expensive hyperparameter tuning, as there are a number of algorithm specific hyperparameters. As optimal hyperparameter settings may vary across implementations and hardware setups, we interpret the results of CAL and SRCPO as strong baseline implementations rather than exhaustively tuned baselines for our experimental setup. The only change made to CAL is to use a 1-1 update-to-data ratio, to isolate its performance under the same amount of collected experience. For both WCSAC and CRAC we use the same hyperparameters for both Safety-Gymnasium and CityLearn.

Table 1: Common Hyperparameters

| Common Hyperparameters | Value |
|---|---|
| Reward Gamma | 0.99 |
| Cost Gamma | 0.99 |
| Policy Lr | $3 \times 10^{-4}$ |
| Reward Critic Lr | $3 \times 10^{-4}$ |
| Cost Critic Lr | $5 \times 10^{-4}$ |
| Agent Entropy AutoTune | True |
| Agent Entropy Lr | $5 \times 10^{-4}$ |
| Tau | 0.005 |
| Critic Net Update Frequency | 5 |
| Actor Net Update Frequency | 10 |
| Lagrangian/PID Update Frequency | 10 |
| Target Net Update Frequency | 10 |
| Buffer Size | $10^6$ |
| Batch Size | 256 |
| Policy Hidden Sizes | [256, 256] |
| Critic Hidden Sizes | [256, 256] |
| Quantile Hidden Sizes | [512, 512] |
| Agent Embedding Dim | 32 |
| Number of Quantiles (Critic) | 32 |
| Number of Quantiles (Policy) | 32 |
| Number of Cost Critics | 2 |
| Learning Starts | 5000 |
| TQC | 1 |
| **CRAC Specific Hyperparameters** | Value |
| Lagrangian MLP Network Size | [64, 64] |
| PID KI | 0.0005 |
| PID KP | 0.1 |
| PID KD | 0.0005 |

Apart from conditioning, the only significant difference between CRAC and WCSAC is that WCSAC uses a dampening factor in the actor update following the Modified Differential Method of Multipliers (Platt & Barr, 1987; Kumar et al., 2021; Yang et al., 2023), in the following form:

$$\bar{\lambda} = \lambda - c(d - Q_C^\pi(s,a)) \tag{16}$$

where $\lambda$ is the Lagrangian multiplier value, $c$ is a dampening parameter, we use a value of 10 for all experiments, $d$ is the cost threshold, and $Q_C^\pi(s,a|\tau)$ is the CVaR cost value estimate. This leads to the actor loss function:

$$J(\pi) = \mathrm{E}_{s \sim \mathcal{B}, a \sim \pi(\cdot|s)}\left[Q_R^\pi(s,a) - \bar{\lambda}\big(Q_{C,\alpha}^\pi(s,a) - d\big) - \beta \log \pi(s,a)\right] \tag{17}$$

where $Q_R^\pi(s,a)$ is the reward value estimate and $\beta$ is the entropy parameter.

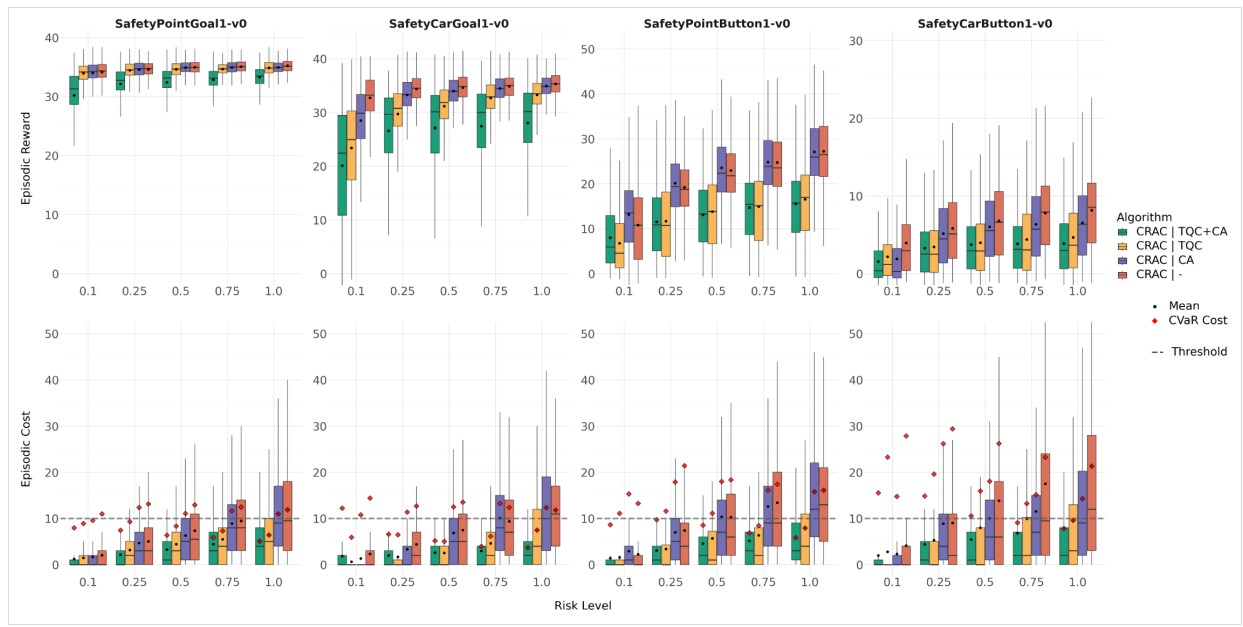

Figure 7: **CRAC** Ablation of TQC and Cost-augmented State. *TQC + CA* - indicates TQC and Cost Augmentation, *TQC* indicates only TQC, *CA* indicates only Cost Augmentation, - indicates neither component is used

# B    Truncated Quantile Critics and Cost Augmented State Ablations

In this section we aim to isolate the effects on WCSAC and CRAC of Truncated Quantile Critics (TQC) and the cost-augmented state (CA), and the effect on SRCPO and CAL of the cost-augmented state.

## B.1    CRAC

Ablation results in Figure 7 show that both TQC and CA improve the safety performance of CRAC. TQC reduces cost underestimation, leading to lower episodic costs and improved CVaR constraint satisfaction across risk levels, while CA further stabilises its CVaR cost satisfaction by providing the policy with necessary information about accumulated cost. Whilst both mechanisms introduce mild conservatism, which leads to reduced reward, together they act as important components in improving the cost performance of CRAC.

## B.2    WCSAC

Ablation results in Figure 8 show similar trends to the ablations for CRAC. TQC improves the safety performance of WCSAC by reducing cost underestimation, leading to lower episodic costs and improved CVaR constraint satisfaction across risk levels. Whilst CA further stabilises the CVaR cost satisfaction of WCSAC by providing the policy with necessary information about accumulated cost.

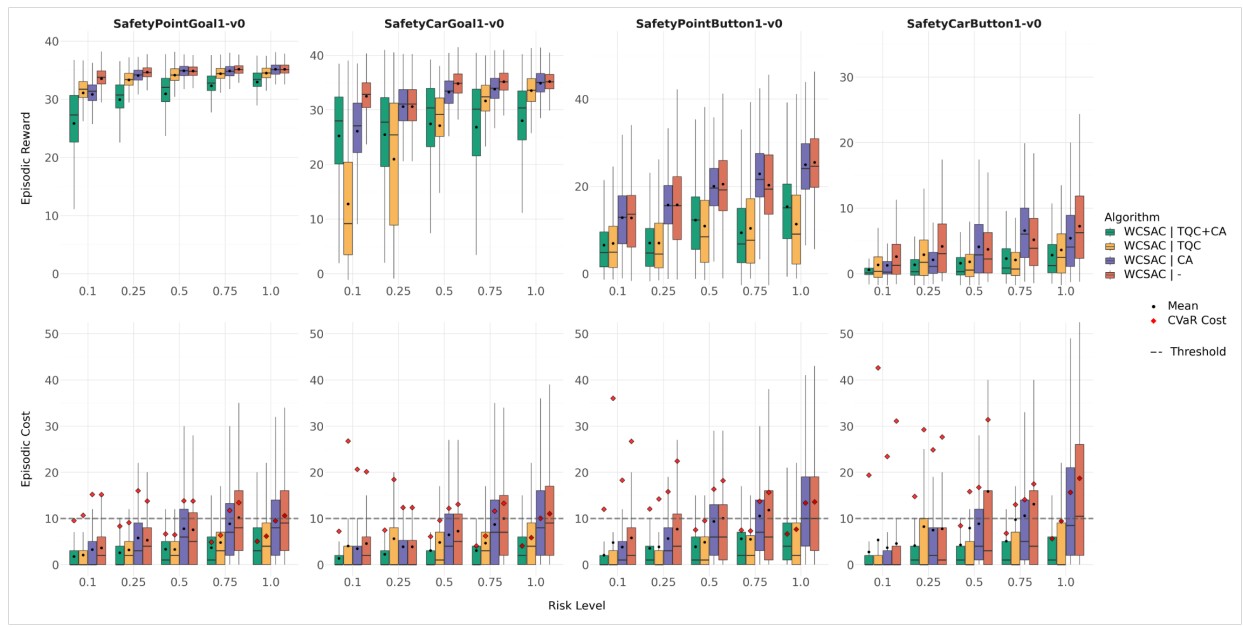

Figure 8: **WCSAC** Ablation of TQC and Cost-augmented State. *TQC + CA* - indicates TQC and Cost Augmentation, *TQC* indicates only TQC, *CA* indicates only Cost Augmentation, - indicates neither component is used

## B.3 SRCPO

Ablation results in Figure 9 show that cost-augmentation has relatively modest effects on the performance of SRCPO. Reward performance is largely similar, with slight improvement in the button tasks. Overall, the impact is less significant to SRCPO than it is to WCSAC and CRAC.

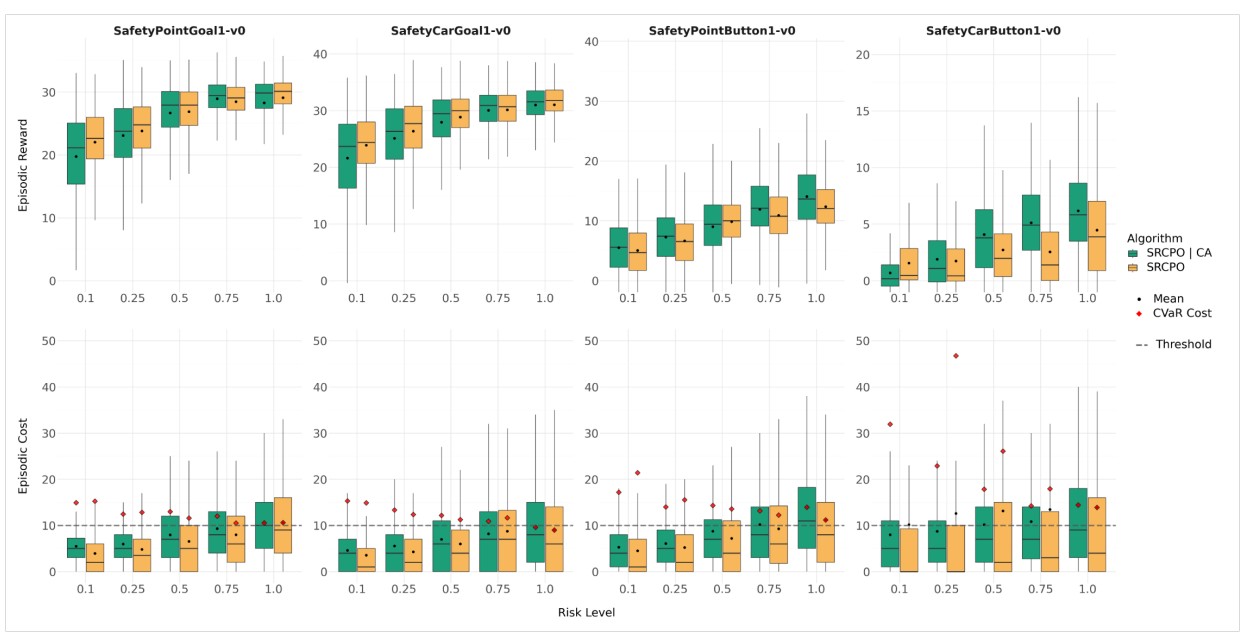

Figure 9: **SRCPO** Ablation of Cost-augmented State

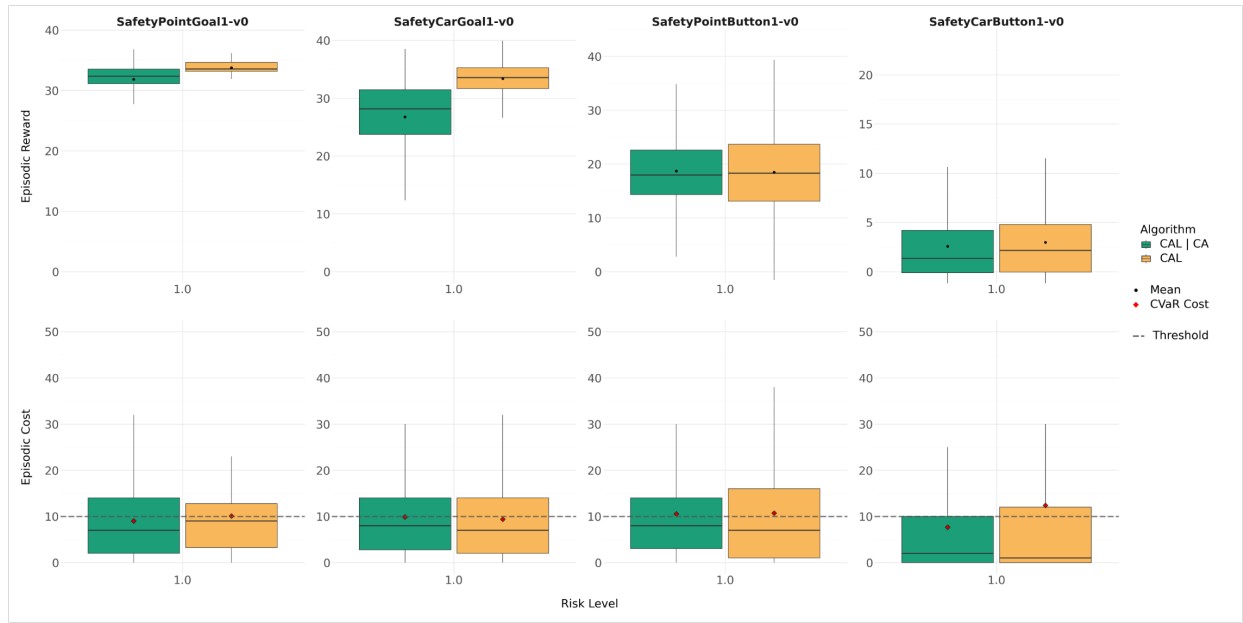

Figure 10: **CAL** Ablation of Cost-augmented State

## B.4   CAL

Ablation results for CAL in Figure 10 show that cost-augmentation slightly reduces the reward performance across the environments, with modest improvements to the cost. It is evident that it induces conservatism that may not be necessary for CAL, providing little benefit to its overall performance. To maintain consistency across the algorithms in our main evaluation, we stick with CAL with state-augmentation, but provide the results for comparison here.

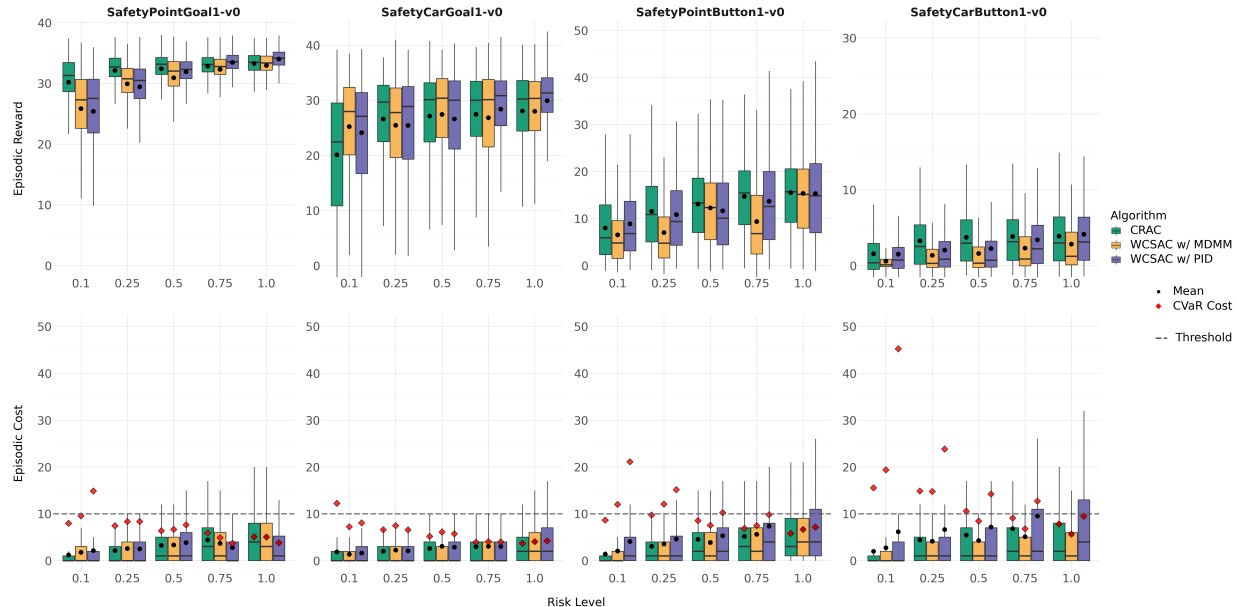

Figure 11: Ablation study of CRAC vs two variants of WCSAC, one with MDMM (WCSAC w/ MDMM) and one with PID (WCSAC w/ PID)

## C  WCSAC w/ PID Ablation

In this section we present results that compare the performance of two variants of WCSAC. Our results presented in the main text made use of WCSAC with the Modified Differential Method of Multipliers (WCSAC w/ MDMM), whereas CRAC made use of a conditional PID-Lagrangian. To better isolate the performance improvements, argued in the main text being largely due to conditioning, we compare both CRAC and WCSAC w/ MDMM against WCSAC with a PID Lagrangian Multiplier (WCSAC w/ PID). For fair comparison, we adopt the same PID hyperparameter values for both WCSAC and CRAC, specifically KI=0.0005, KP=0.1, KD=0.0005.

From the results presented in Figure 11, we can observe that in the Goal environments, performance is largely comparable at the more risk-averse risk levels, with WCSAC w/ PID failing to satisfy the CVaR constraint at $\alpha = 0.1$ in PointGoal1. At more risk-neutral levels, WCSAC w/ PID outperforms WCSAC w/ MDMM and CRAC. However, in the Button environments, WCSAC w/ PID largely fails to satisfy the CVaR constraints. There are clear signs of reward improvement when paired with the PID, but at the expense of cost performance, indicating that WCSAC w/ MDMM leads to better constraint satisfying policies.

In summary, these results, paired with the risk level sampling ablations in Section 5.3.1, further aid in isolating the performance improvements of CRAC, indicating that conditioning and uniform relabelling play a key role.

## D  Computational Complexity of Algorithms

All experiments were conducted on a PC with an AMD EPYC 7763 CPU processor and a NVIDIA-A100 80GB GPU. In Table 2, we report the training times for each algorithm in the Safety Point Goal environment. The significant difference between CRAC and WCSAC comes largely from two CRAC specific components. Firstly, the training of the Conditional PID Lagrangian, represented as two [64,64] layer neural networks, is more computationally expensive than the single Lagrangian parameter in WCSAC. On top of this, we also carry out in-training evaluations for all algorithms, every 100k steps, we evaluate each algorithm for 20 episodes. Where the fixed-risk algorithms, and CAL, evaluate a single policy every 100k steps, for CRAC we evaluate across a set of risk levels, $\alpha \in \{0.1, 0.25, 0.5, 0.75, 1.0\}$, that aligns with the risk levels WCSAC

and SRCPO are trained at. It is worth noting that the longer training time of SRCPO may not necessarily be due to increased computational complexity over the other algorithms, rather its implementation which we use from its original repository, detailed in Section A.

In the Safety-Gymnasium PointGoal1 environment, training one CRAC agent takes on average 5 hrs and 25 minutes, whereas training 5 WCSAC agents at these risk levels takes on average 19 hrs. Therefore, the increase in training time of one CRAC agent vs one WCSAC, shown in Table 2, is small relative to the training time that is required to train multiple WCSAC agents at these 5 risk levels.

Table 2: Training times in Point Goal Environment, averages across five seeds

| - | CRAC (OURS) | WCSAC | SRCPO | CAL |
|---|---|---|---|---|
| TRAINING TIME | 5H 25M | 3H 50 M | 17H 30M | 5H 10M |

# E    Environment Specific Figures

In the main text, in Figure 1, the results in Safety-Gymnasium comparing CRAC against the set of baselines are plotted in box plots, across multiple risk levels and multiple environments, which may lead to difficulty in interpreting the results, particularly at specific risk levels. In light of this, this section presents the result figures for each environment.

## E.1    PointGoal1

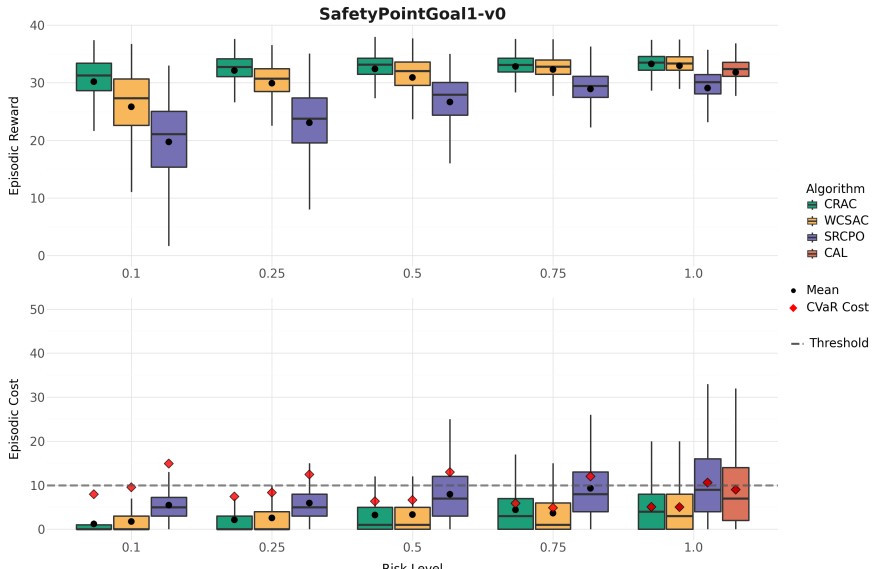

Figure 12: **PointGoal Reward-Cost Distributions.** CRAC achieves the highest reward at lower risk levels while WCSAC becomes more competitive towards higher risk levels. SRCPO fails to satisfy CVaR constraints and achieves the lowest reward

## E.2   CarGoal1

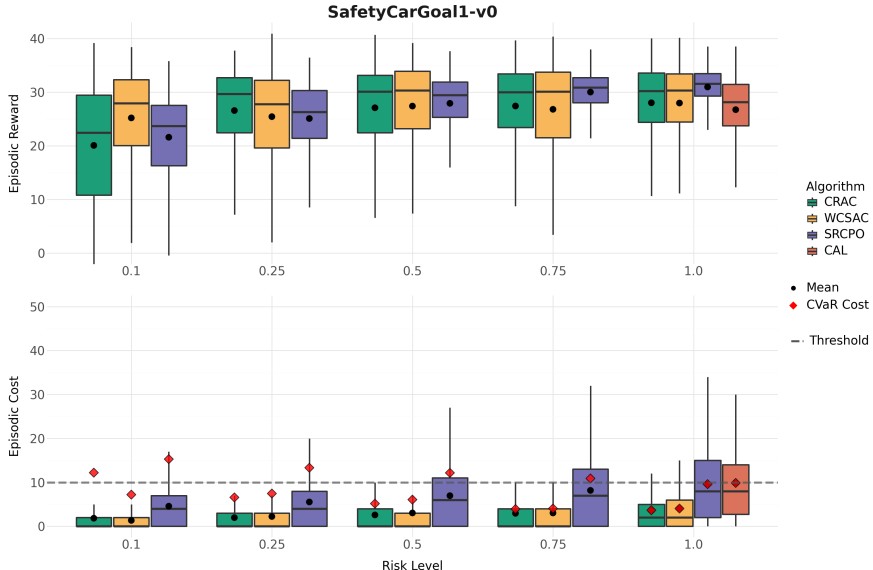

Figure 13: **CarGoal1 Reward-Cost Distributions.** CRAC violates the constraint at the lowest risk level, and achieves the lowest reward, becoming more competitive with WCSAC towards higher risk levels. SRCPO fails to satisfy CVaR constraints but is competitive in reward with both CRAC and WCSAC, achieving the best risk-neutral policy.

## E.3   PointButton1

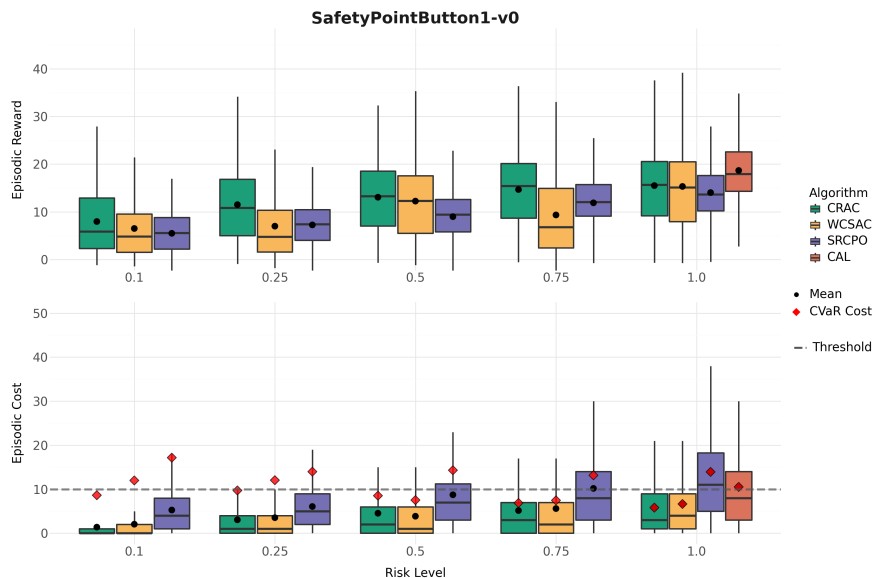

Figure 14: **PointButton1 Reward-Cost Distributions.** CRAC achieves the highest reward across all risk levels, with WCSAC being competitive at $\alpha \in \{0.5, 1.0\}$. SRCPO fails to satisfy CVaR constraints but is competitive in reward with WCSAC.

### E.4 CarButton1

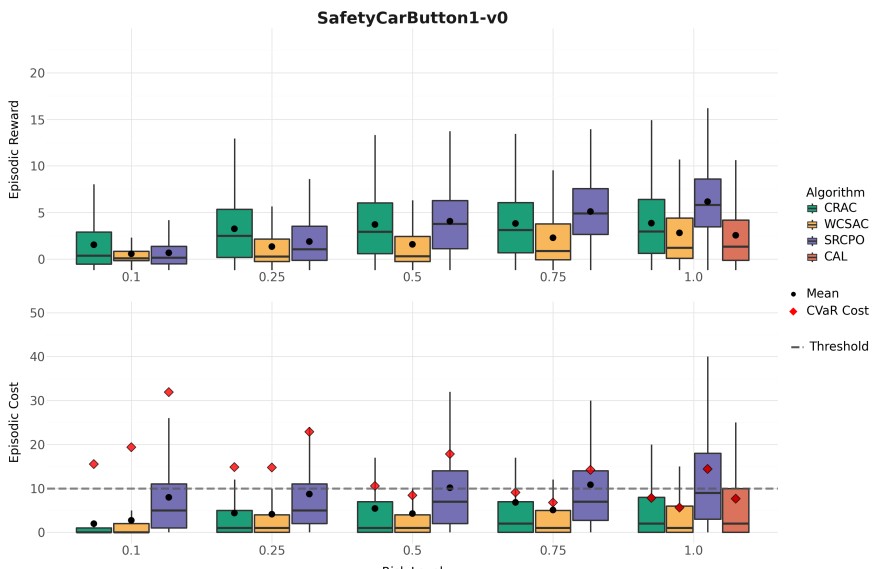

Figure 15: **CarButton1 Reward-Cost Distributions.** We note that all algorithms fail to satisfy the CVaR based constraints at the lower risk levels $\alpha \in [0.1, 0.25]$, whilst exhibiting a low distribution of costs. This indicates a common challenge in the Safety-Gymnasium environments of rare but high cost episodes (Ji et al., 2023). However, as satisfying CVaR constraints becomes increasingly more difficult as $\alpha$ tends towards more risk-averse values, constraint satisfaction may become near-infeasible at more risk-averse levels, under a fixed constraint.

