# OpenReview forum: "Conditional Risk-Averse Constrained Reinforcement Learning"
_TMLR — Accepted by TMLR_

### Review · Reviewer_BNQW · 2026-04-02

**Summary Of Contributions:**

The authors propose Conditional Risk-Averse Actor Critic (CRAC), a SAC-based actor–critic with IQN cost critics, augmented by conditioning the policy, reward Q-networks, cost quantile field, and dual variable on $\alpha$. Evaluations cover SafetyGymnasium and CityLearn and are compared to WCSAC, SRCPO, and CAL. There are also truncated-$\alpha$ generalization tests and ablations on $\alpha$ sampling and PID multipliers.

Strengths

- Learning one policy $\pi(\cdot \mid s, \alpha)$ across risk levels rather than retraining per preference is a practical and useful idea. The paper presents the components clearly and ties them to the CMDP formulation.
- The $\alpha$-sampling ablation is informative — uniform resampling helps cost satisfaction, and the connection to implicit cost overestimation is an interesting observation.
- CityLearn, generalization experiments and OOD evaluation are thorough.

Weaknesses/comments:

- As CVaR depends on the policy, $\alpha$ relabeling could be problematic as it changes the CVaR computation, the multiplier $\lambda(\alpha)$, and the Bellman target policy. Is there any analysis of what distribution the critics converge to under this approach? Similarly, the cost-augmented state $e_t$ was accumulated under $\pi(\cdot \mid s, \alpha_{\mathrm{e}})$. After relabeling with $\alpha_{\mathrm{tr}}$, $e_t$ still reflects $\alpha_{\mathrm{e}}$'s cost history. Maybe I missed something, but this seems like a mismatch the critic would have to handle.

- While authors mention that WCSAC uses MDMM and CRAC uses PID, Section 5.3.2 partially addresses this. But it's still CRAC-with-I, not WCSAC-with-PID. The dual mechanism difference isn't fully isolated from the conditioning effect.

- Using fixed $d$ across all $\alpha$: since CVaR is monotone in $\alpha$, the chosen threshold could be near-infeasible at very low $\alpha$. The authors attribute low-$\alpha$ violations to rare high-cost episodes, which is reasonable, but I think a brief feasibility discussion as a function of $\alpha$ is missing.

- Since there's no theoretical analysis, I also think adding wall-clock or GPU-hour comparison for CRAC vs training $N$ separate WCSAC agents would help.

- CCPO conditions a safe RL policy on the constraint threshold $d$ which is still conceptually very similar. I think a comparison, or a more detailed discussion of why $\alpha$-conditioning in the risk-averse setting poses additional challenges, would help.
- In the box plots, the differences are hard to interpret due to high variance (five seeds, many $\alpha$ values, no confidence intervals).
- OOD behaviour in $\alpha$ could be stated more explicitly as a limitation in the main text, not only via figures.

**Audience:**

Yes

**Audience Explanation:**

The work is important for researchers in safe RL, constrained MDPs, and distributional representations for risk-sensitive safety constraints.

**Claims And Evidence:**

Yes

**Claims Explanation:**

The algorithm and experiments support the main empirical story and the method is sufficiently detailed.

**Requested Changes:**

Please address the weaknesses mentioned in the Summary section.

---

> ### Author Response · Authors · 2026-04-29
> **Response to Reviewer BNQW (part 1)**
>
> We firstly want to thank the reviewer for taking the time to read and review our paper and for providing value feedback.
>
> > As CVaR depends on the policy, $\alpha$ relabelling could be problematic as it changes the CVaR computation, the multiplier $\lambda(\alpha)$, and the Bellman target policy. Is there any analysis of what distribution the critics converge to under this approach? Similarly, the cost-augmented state $e_{t}$ was accumulated under $\pi(\cdot|s,\alpha_{e})$. After relabelling with $\alpha_{tr}$, $e_{t}$ still reflects $\alpha_{e}$'s cost history. Maybe I missed something, but this seems like a mismatch the critic would have to handle.
>
> While we have not provided a direct analysis of the convergence of the critics under $\alpha$ relabelling, the included risk-level sampling ablations aims to empirically examine the effect of this mismatch, by comparing CRAC under different relabelling approaches, uniform and local, and no relabelling (stored). These results in particular suggest that relabelling can improve the reward-cost trade-off of CRAC, but its effect depends on the environment and relabelling approach.
>
> Regarding the cost-augmented state $e_{t}$, we agree that this reflects the cost history generated under the behavioural policy $\alpha_e$. However, $e_{t}$ retains valid information about a trajectory, measuring the accumulated cost up to that state. After relabelling, it, in effect, becomes an additional conditioning variable for the actor and critic. As different policies may visit similar states having accumulated different costs, $e_{t}$ allows the networks to distinguish between these scenarios and may encourage more conservative action selection when accumulated cost is already high. The results presented in the appendix show that cost augmentation improves CRAC in a similar manner to WCSAC, suggesting that the actor and critic make effective use of this information despite the mismatch.
>
> However, while empirically CRAC's performance is improved by both relabelling and by using a cost-augmented state, a formal analysis of both their effects on the convergence of the critic remains important future work.
>
>
> > While authors mention that WCSAC uses MDMM and CRAC uses PID, Section 5.3.2 partially addresses this. But it's still CRAC-with-I, not WCSAC-with-PID. The dual mechanism difference isn't fully isolated from the conditioning effect.
>
> We acknowledge that the dual mechanism and conditioning effect are not fully isolated in our results. We have added ablation results with WCSAC w/ PID to the appendix, Section C, to better isolate the conditioning effect from WCSAC. In summary, the results suggest that while WCSAC w/ PID outperforms WCSAC w/ MDMM in terms of reward in the Button environments, it comes at the expense of poorer cost performance as WCSAC w/ PID violates the constraints at more risk-averse levels. Further to this, in the PointGoal environments, WCSAC w/ PID slightly under performs WCSAC w/ MDMM in terms of reward at more risk-averse levels, as well as violating the constraint at the most risk averse level in PointGoal. We believe these results, paired with the results studying the effect of risk level sampling in Section 5.3.1, better isolate the performance improvement of CRAC vs WCSAC to the conditioning on risk levels.
>
> > Using fixed $d$ across all $\alpha$: since CVaR is monotone in $\alpha$, the chosen threshold could be near-infeasible at very low $\alpha$. The authors attribute low-$\alpha$ violations to rare high-cost episodes, which is reasonable, but I think a brief feasibility discussion as a function of $\alpha$ is missing.
>
> We thank the reviewer for this suggestion and agree that a discussion of the feasibility of a fixed threshold with respect to $\alpha$ is necessary. We have added a brief discussion addressing this to our results section, clarifying that the observed constraint violations may be reflective of the constraint approaching near-infeasibility, rather than solely due to rare high-cost episodes.

---

> ### Author Response · Authors · 2026-04-29
> **Response to Reviewer BNQW (part 2)**
>
> > Since there's no theoretical analysis, I also think adding wall-clock or GPU-hour comparison for CRAC vs training separate WCSAC agents would help.
>
> We thank the reviewer for this suggestion and in the appendix, Section D, we have added a comparison of the time taken to train each agent in the Safety Gymnasium PointGoal1 environment.
> In practice, there are two main places in which CRAC's training time would increase over WCSAC.
> (i) In training the conditional Lagrangian multiplier rather than a single parameter Lagrangian multiplier, and (ii) during training we evaluate CRAC for 20 episodes every 100K timesteps at the risk levels [0.1,0.25,0.5,0.75,1.0] (i.e. the same risk levels we train the fixed WCSAC). This is largely to monitor the performance of CRAC during training and can be adjusted depending on the preference of the user. In practice this would be the most significant training time increase, as rather than evaluating one policy during the course of training CRAC evaluates 5 (at least in our setup).
>
> > CCPO conditions a safe RL policy on the constraint threshold $d$ which is still conceptually very similar. I think a comparison, or a more detailed discussion of why $\alpha$-conditioning in the risk-averse setting poses additional challenges, would help.
>
> We agree that conditioning on the constraint and on $\alpha$ are conceptually similar, we have revised the Related Work section to better distinguish CCPO and CRAC. Specifically, introducing conditional risk-aversion requires approximations of the cost return distribution under different policies and evaluating risk measures of the return distribution across risk levels. CRAC makes use of distributional value functions and a conditional Lagrangian multiplier to support policy updates under CVaR constraints. In contrast, CCPO learns expected-cost critics and updates the policy through Expectation-Maximisation. Specifically, in CCPO threshold conditioning changes the allowable expected cost threshold, whereas in CRAC $\alpha$ conditioning changes which part of the cost-return distribution the constraint corresponds to.
>
> > In the box plots, the differences are hard to interpret due to high variance (five seeds, many $\alpha$ values, no confidence intervals).
>
> We agree that differences can be difficult to interpret due to the many risk levels and multiple environments in a single graph. The intention behind using box plots was to present the distribution of performance across seeds and risk levels, highlighting the broader trend of reward improvement in CRAC, rather than emphasising single measures at each individual risk level. We have since increased the size of the overlaid the mean episodic reward and cost (black points), as well as the CVaR cost (red diamonds). As well as this we have added to the appendix, Section E, environment specific box plots, to better separate the plotted results.
>
> > OOD behaviour in $\alpha$ could be stated more explicitly as a limitation in the main text, not only via figures.
>
> We have restated the discussion of OOD behaviour to make it more explicit that CRAC's generalisation to OOD risk levels is limited to risk levels close to its training distribution.

---

### Review · Reviewer_qhWA · 2026-04-06

**Summary Of Contributions:**

This work proposes a practical algorithm for constrained reinforcement learning, in the case in which the constraint is expressed as a risk measure instead of an expectation. While algorithms such as WCSAC have already been proposed for this setting, this algorithm explicitly conditions the policy, critics and Lagrangian multiplier on the risk level $\alpha$, which can then sampled both during training and inference. The result agent is thus adaptive: the risk level can be controlled directly during inference. Furthermore, the authors integrate a common PID scheme to stabilize the Lagrangian multiplier.

The resulting algorithm is evaluated across standard SafetyGym environments and CityLearn: while performance gains are modest, the algorithm compares favorably against baselines. The main results are correlated by a sequence of ablations studying (i) generalization to unseen risk levels (which is enabled by the method) (ii) risk level sampling techniques (displaying that resampling risk levels is beneficial) and (iii) the effectiveness of PID components.

**Strengths:**
- This submission is well written and clearly presented.
- The experimental evaluation is thorough, and the selection of environments is sufficient to support the claims.
- The evaluation of OOD risk revels, showing a degree of generalization, is particularly interesting.

**Weaknesses:**
- To the best of my understanding, the new method adopts two main changes with respect to WCSAC: it is conditional on risk levels, and it integrates a PID component. While the performance of the method is attributed entirely to the first element, it is not clear whether this is the case. Is WCSAC also using same the PID component? Only in this case this conclusion is well-supported.

**Further comments:**
- The novelty of the method is questionable: conditional risk-averseness has been explored in non-constrained settings; the adaptation to CRL involves standard techniques and does not contribute meaningful insights. Given the guidelines of TMLR, this does not represent a weaknesses per se, but I would like to ask the authors why their submission goes beyond a direct combination of WCSAC and RC-DSAC.
- The last paragraph of Section 4 is largely redundant, as it summarizes contributions that were just explained clearly.

**Additional Comments:**

Constrained and risk-averse RL are not my main research field, although I have worked on related topics. It is possible that I am unaware of significantly related works. Please, take into account a medium-to-low confidence on my evaluation.

**Audience:**

Yes

**Audience Explanation:**

While, to the best of my understanding, this submission is a direct combination of WCSAC with RC-DSAC, this specific combination was not published in the past, and it therefore may be of interest to the community.

**Broader Impact Concerns:**

None.

**Claims And Evidence:**

Yes

**Claims Explanation:**

In my opinion, the main issue of this submission is the attribution of performance gain to conditional risk levels, when other components (namely the PID one) are also introduced.

Edit: this was ablated by the authors during the rebuttal.

**Requested Changes:**

The main requested changes is a detailed ablation for attributing the performance improvements to either the variable risk level or the PID component. I would also like to ask the authors to answer my questions in "Further Comments", but this is not necessary for acceptance (it might still strengthen the work, however).

---

> ### Author Response · Authors · 2026-04-29
> **Response to Reviewer qhWA**
>
> We wish to thank the reviewer for taking the time to review our work and for providing value feedback and suggestions.
>
> > To the best of my understanding, the new method adopts two main changes with respect to WCSAC: it is conditional on risk levels, and it integrates a PID component. While the performance of the method is attributed entirely to the first element, it is not clear whether this is the case. Is WCSAC also using same the PID component? Only in this case this conclusion is well-supported.
>
> > In my opinion, the main issue of this submission is the attribution of performance gain to conditional risk levels, when other components (namely the PID one) are also introduced.
>
> > The main requested changes is a detailed ablation for attributing the performance improvements to either the variable risk level or the PID component.
>
> We acknowledge this concern raised by reviewer qhWA, which is similar to that raised by BNQW. For readability, we repeat the same answer here as provided to reviewer BNQW. We acknowledge that the dual mechanism and conditioning effect are not fully isolated in our results, as WCSAC does not use the PID component. We have added ablation results with WCSAC w/ PID to the appendix, Section C, to better isolate the conditioning effect from WCSAC. In summary, the results suggest that while WCSAC w/ PID outperforms WCSAC w/ MDMM in terms of reward in the Button environments, it comes at the expense of poorer cost performance as WCSAC w/ PID violates the constraints at more risk-averse levels. Further to this, in the PointGoal environments, WCSAC w/ PID slightly under performs WCSAC w/ MDMM in terms of reward at more risk-averse levels, as well as violating the constraint at the most risk averse level in PointGoal. We believe these results, paired with the results studying the effect of risk level sampling in Section 5.3.1, better isolate the performance improvement of CRAC vs WCSAC to the conditioning on risk levels.
>
> > The novelty of the method is questionable: conditional risk-averseness has been explored in non-constrained settings; the adaptation to CRL involves standard techniques and does not contribute meaningful insights. Given the guidelines of TMLR, this does not represent a weaknesses per se, but I would like to ask the authors why their submission goes beyond a direct combination of WCSAC and RC-DSAC.
>
> While CRAC shares similarities with RC-DSAC, as both approaches condition policies on risk levels, we argue that a direct combination of WCSAC and RC-DSAC is non-trivial, and that the main novelty of CRAC is the formulation and empirical analysis of conditional risk-aversion in the constrained setting.
>
> This extension to the constrained setting introduces challenges not present in the prior works. Specifically, CRAC contributes beyond a direct combination through: (i) a Conditional PID-Lagrangian, motivated by the instability common to Lagrangian-based methods, and exacerbated by adapting the multiplier to a range of risk levels, with the ablations in Section 5.3.2 suggesting improved stability, particularly at lower risk levels; (ii) Risk level sampling and relabelling ablations, which study the effect of $\alpha$ relabelling on the performance of CRAC, particularly its effect on constraint satisfaction, and (iii) an evaluation of out-of-distribution (OOD) generalisation, which is particularly relevant for the CRL setting, as poor generalisation to OOD risk levels can lead to constraint violations.
> We therefore believe that the novelty of CRAC lies not only in combing policy-conditioning and risk-constrained RL, but in identifying and empirically studying the stability, relabelling and the generalisation challenges that arise in the CRL setting.
>
> > The last paragraph of Section 4 is largely redundant, as it summarizes contributions that were just explained clearly.
>
> We thank the reviewer for this suggestion and have removed this paragraph for improved clarity.

---

> > ### Comment · Reviewer_qhWA · 2026-04-29
> >
> > Thank you for the rebuttal. My constraints are addressed, and I think that the contribution of conditioning is now better disentangled. The fact that a good chunk of the performance of the method comes from the PID component should however be further highlighted in the main body of the paper.

---

> > > ### Author Response · Authors · 2026-04-30
> > > **Response to Reviewer qhWA**
> > >
> > > Thank you for this suggestion. We have revised the main text, specifically the Conditional Lagrangian Multiplier ablation results 5.3.2 and the Conclusion,  to more clearly highlight CRAC’s performance improvements arises from both risk-conditioning and improved stability from the PID Lagrangian update, rather than conditioning alone.

---

### Review · Reviewer_kVhj · 2026-04-22

**Summary Of Contributions:**

Risk aware Constrained RL (CaCRL) is RL but with additional safety constraints embedded into the training process. This introduces an apparent tradeoff between reward and safety, and uncertainty further compounds this tradeoff. Often the acceptable risk is fixed once and then an agent is trained. In contrast, the authors propose a method CRAC which allows conditioning on the risk level, observed during training. The authors show experimentally that the method works across different risk levels. No

**Additional Comments:**

**Strengths:**
- The paper starts by reviewing related literature. I am not an expert in this area but the discussion has a good mix of old and new references and gives a sense of the area.
- I believe the core problem (navigating the risk-reward tradeoff, and in fact even generalising better when considering multiple risk levels) is clearly motivated and solves a relevant practical problem.
- I believe the claims are suitable. The authors don't claim uniform improvement over all methods, which is sensible.

**Weaknesses:**
- In comparing with other methods, it is difficult to understand the fairness given to each method.
    - In Appendix A, it is stated that "We reimplement CAL, making use of the official hyper-parameters used for CAL". Two questions: (1) Why did you reimplement it (did the official code simply not work?) and (2) Rather than using a fixed set of hyperparameters, regardless of whether they are "official" or not, wouldn't it be more appropriate to allocate the same hyperparameter tuning budget to each method?
    - " and we use the official code repositiory for SRCPO (https://github.com/rllab-snu/Spectral-Risk-Constrained-RL)". Hyperparameter tuning of this method is not discussed. Did you do a hyperparameter search?
- It is challenging to find the computational time for each method. Did you report these?

**Audience:**

Yes

**Audience Explanation:**

Risk-aware RL is a very important topic in safety. Uncertainty quantification more broadly is of interest to the machine learning community. I have no doubts that some readers will be interested in this paper.

**Broader Impact Concerns:**

No concerns.

**Claims And Evidence:**

Yes

**Claims Explanation:**

A total of 6 figures detailing experimental performance are provided showing a range of problems and comparison to other methods. A mathematical derivation is provided for the proposed new algorithm. I am not knowledgeable in RL but it seems reasonable. I would encourage the authors to provide more details about the hyperparameter selection and computation time; see detailed comments below.

**Requested Changes:**

I believe this paper is already strong. That being said, there are a few changes which I believe will strengthen the paper and make it more interesting for readers, eventually leading to higher impact. See weaknesses below for details. In order of importance:
1. I strongly recommend the authors to detail the computational cost for each of the methods, if only on a subset of problems
2. Be more precise about reporting hyperparameter tuning (and ideally report the time taken here too, hopefully it is about the same for each method).

Minor typos:
- "trade-off's" should be "trade-offs"
- I think "auxillary" is mispelled
- "paramterised" typo
- "spectram" should be "spectrum"
- "pervious"
- In appendix A, "repisitory"

There are numerous awkward grammatical constructions, try and fix these.

---

> ### Author Response · Authors · 2026-04-29
> **Response to Reviewer kVhj**
>
> We want to thank the reviewer for taking the time review our work and providing value feedback. Firstly, we have corrected the typos highlighted and will continue to proof read the paper to further improve future revisions/camera-ready.
>
> > I strongly recommend the authors to detail the computational cost for each of the methods, if only on a subset of problems
> It is challenging to find the computational time for each method. Did you report these?
>
> We thank the reviewer kVhj for this recommendation, a similar recommendation has been made by reviewer BNQW, so for readability, we repeat the same answer here, as provided to reviewer BNQW. In the appendix we have added a comparison of the time taken to train each agent in the Safety Gymnasium PointGoal1 environment.
> In practice, there are two main places in which CRAC's training time would increase over WCSAC.
> (i) In training the conditional Lagrangian multiplier rather than a single parameter Lagrangian multiplier, and (ii) during training we evaluate CRAC for 20 episodes every 100K timesteps at the risk levels [0.1,0.25,0.5,0.75,1.0] (i.e. the same risk levels we train the fixed WCSAC). This is largely to monitor the performance of CRAC during training and can be adjusted depending on the preference of the user. In practice this would be the most significant training time increase, as rather than evaluating one policy during the course of training CRAC evaluates 5 (at least in our setup).
>
>
> > Be more precise about reporting hyperparameter tuning (and ideally report the time taken here too, hopefully it is about the same for each method).
>
> > In comparing with other methods, it is difficult to understand the fairness given to each method.
> In Appendix A, it is stated that "We reimplement CAL, making use of the official hyper-parameters used for CAL". Two questions: (1) Why did you reimplement it (did the official code simply not work?) and (2) Rather than using a fixed set of hyperparameters, regardless of whether they are "official" or not, wouldn't it be more appropriate to allocate the same hyperparameter tuning budget to each method?
> " and we use the official code repositiory for SRCPO (https://github.com/rllab-snu/Spectral-Risk-Constrained-RL)". Hyperparameter tuning of this method is not discussed. Did you do a hyperparameter search?
>
> With respect to the hyperparameter discussion in the appendix, we have revised it to better clarify the reasoning behind our hyperparameter choices.
>
> Answering question (1): We reimplemented CAL largely in order to better integrate with our training setup and in order to gain training time speed ups.
>
> Answering question (2, part 1): We adopted CAL's hyperparameters to ensure our results accurately reflected the algorithm's best performance levels found by CAL's authors. In their paper the authors performed a number of ablations on the key hyperpameters they introduced (i.e. Critic Ensemble size, Critic Upper Confidence Bound and the Augmented Lagrangian scaling parameter), which we adopted rather than performing a time-consuming hyperparameter search on these parameters. A number of other hyperparameters values are values typically used in other works, as well as common to WCSAC and CRAC in our work (such as network size and learning rates), to as fairly compare the methods as possible. To the best of our knowledge this is standard practice in RL as to best isolate the added algorithmic changes of the proposed method, rather than attribute any chances to specific hyperparameter tuning.
>
> Answering question (2, part 2): Regarding SRCPO, we did not perform hyperparameter search, as SRCPO is quite time-consuming to train and has a number of algorithm specific hyperparameters, therefore we adopted the original implementations hyperparameters, under the assumption that they would lead to the highest performance of the algorithm.

---

> > ### Comment · Reviewer_kVhj · 2026-05-11
> >
> > Thanks for your response. I will maintain my positive evaluation.
> >
> > I think your revision could be a little bit more open and critical in acknowledging the limitation of your hyperparameter selection. According to your rebuttal, you make two key choices:
> > 1. Reimplement CAL
> > 2. Use the hyperparameters that reportedly gave the best performance in a previous implementation.
> >
> > These two choices taken together, in my experience, will not result in an optimal hyperparameter choice for your reimplementation. This is because the complexity of the algorithms, data, randomness, hardware etc. can result in quite specific hyperparameter optimisations. I am not expecting you to change your setup, just to acknowledge this as a limitation of your experiment.
> >
> > Regarding question (2, part 2), please also highlight this limitation.

---

> > > ### Author Response · Authors · 2026-05-26
> > > **Response to Reviewer kVhj**
> > >
> > > We thank the reviewer for this suggestion and agree that this limitation should be highlighted more explicitly in our paper. We have revised Section 5.1.2 where we discuss the comparison algorithms in our experiment to acknowledge the potential for the performance of reimplemented baselines to vary due to differences in experimental setups such as randomness, hardware and implementation details. Specifically, we clarify that CAL and SRCPO should be interpreted as strong baseline implementations under the reported hyperparameter settings, rather than exhaustively tuned baselines.

---

### Author Response · Authors · 2026-04-29
**Revision 1 Description**

Firstly we would like to thank the reviewers for their time and effort in reviewing our paper, and for the useful feedback and suggestions. We have made a revision to the paper incorporating the feedback and suggested changes.

 Specifically:

1. We have added an ablation study to better isolate the performance improvements of CRAC relative to WCSAC, by comparing CRAC against two variants of WCSAC, one with MDMM and one with PID, in the Safety Gymnasium environments. This is presented in the Appendix, Section C.
2. We have added a table to the appendix presenting the computational complexity of each algorithm in the Safety Gymnasium Point Goal1 environment. This is presented in the Appendix, Section D.
3. We have made small visual changes to figure presenting the main set of results in the Safety Gymnasium to try to improve its visual clarity, as well as including environment specific box plot figures in the Appendix, in Section E.
4. We have revised the Related Work section to better distinguish between risk-conditioning and constraint-conditioning,
5. We have included a brief discussion of the fixed constraint threshold becoming near-infeasible at more risk-averse levels.
6. We have restated how OOD generalisation is limited to risk levels closer to the training risk distribution.
7. We have clarified the hyperparameter choices of each algorithm in the Appendix Section A.
8. Lastly we have fixed the spelling mistakes highlighted by reviewer kVhj, and will continue to proof read to catch further spelling or grammar mistakes, and correct in a further revision/camera-ready.

---

> ### Author Response · Authors · 2026-04-30
> **Revision 2 Description**
>
> We have revised the main text in response Reviewer qhWA's suggestion to more clearly highlight that CRAC’s performance improvements arise from both risk-conditioning and improved stability from the PID Lagrangian update, rather than conditioning alone. Specifically, we have revised the text in the Conditional Lagrangian Multiplier ablation results (Section 5.3.2) and the Conclusion.

---

> > ### Author Response · Authors · 2026-05-26
> > **Revision 3**
> >
> > We revised the paper to more explicitly acknowledge the limitations associated with our reimplementation of CAL using reported hyperparameters, and our use of SRCPO with its reported hyperparameters. Specifically in Section 5.1.2 we acknowledge this limitation.
> > We have also corrected two typos, (i) the incorrect indexing of action $a_{t+1}$ after E.q 11, and (ii) the incorrect nesting of the exploration phase (Phase 1) and gradient update phase (Phase 2) in Algorithm 1.

---

### Decision · Action_Editor_zujU · 2026-06-08

**Recommendation:** Accept as is

**Audience:**

Yes

**Audience Explanation:**

Risk-aware RL is a very important topic in safety. The work is definitely of value to researchers and practitioners in safe RL, constrained MDPs, and distributional representations for risk-sensitive safety constraints.

**Claims And Evidence:**

Yes

**Claims Explanation:**

All reviewers agreed that the work constitutes a valid scientific contribution with improved ablations that separate the impact of the variable risk level vs. the PID component. They also found the rebuttal adequate in terms of empirical comparison with other methods and hyperparameter selection. The authors additionally addressed the AE's concerns. Strengths:
- Problem is well-motivated and interesting / relevant to RL practitioners.
- Theory seems correct, as far as I can tell.
- Extensive literature review with very good coverage.
- This submission is well written and clearly presented.
- The experimental evaluation is very thorough, and the selection of environments is sufficient to support the claims.
- The evaluation of OOD risk revels, showing a degree of generalization, is particularly interesting.

The authors are still encouraged to carefully proofread their manuscript for "hidden" typos or errors that the Reviewers and the AE missed.